# REOBench: Benchmarking Robustness of Earth Observation Foundation Models

**Xiang Li**[1][*]**, Yong Tao**[2][*]**, Siyuan Zhang**[3][*]**, Siwei Liu**[4]**, Zhitong Xiong**[5]
**Chunbo Luo**[2]**, Lu Liu**[2]**, Mykola Pechenizkiy**[6]**, Xiao Xiang Zhu**[5]**, Tianjin Huang**[2,6][†]

[1] University of Bristol, UK [2] University of Exeter, UK
[3] South China Normal University, China [4] The University of Aberdeen, UK
[5] Technical University of Munich, Germany [6] Eindhoven University of Technology, NL

## Abstract

Earth observation foundation models have shown strong generalization across multiple Earth observation tasks, but their robustness under real-world perturbations remains underexplored. To bridge this gap, we introduce REOBench, the first comprehensive benchmark for evaluating the robustness of Earth observation foundation models across six tasks and twelve types of image corruptions, including both appearance-based and geometric perturbations. To ensure realistic and fine-grained evaluation, our benchmark focuses on high-resolution optical remote sensing images, which are widely used in critical applications such as urban planning and disaster response. We conduct a systematic evaluation of a broad range of models trained using masked image modeling, contrastive learning, and vision-language pre-training paradigms. Our results reveal that ❶ existing Earth observation foundation models experience significant performance degradation when exposed to input corruptions. ❷ The severity of degradation varies across tasks, model architectures, backbone sizes, and types of corruption, with performance drop varying from less than 1% to over 25%. ❸ Vision-language models show enhanced robustness, particularly in multimodal tasks. REOBench underscores the vulnerability of current Earth observation foundation models to real-world corruptions and provides actionable insights for developing more robust and reliable models. Code and data are publicly available at https://github.com/lx709/REOBench.

## 1 Introduction

Recent studies have shown that foundation models pre-trained on large-scale datasets have demonstrated powerful capabilities across multiple domains. Models such as MAE [1], CLIP [2], MiniGPT-4 [3], and LLaVA [4] have achieved remarkable success in multiple vision and vision-language tasks. These models are capable of extracting representative features from images, and more importantly, can be quickly adapted to multiple downstream tasks with minimal fine-tuning, significantly improving the efficiency and effectiveness of task-solving.

In the field of remote sensing, the rapid growth in data volume has sparked significant interest in developing foundation models for remote sensing image analysis. Earth observation foundation models (EOFMs) aim to leverage supervised or self-supervised training to build large-scale pretrained models that can be adapted to a wide range of downstream tasks, thereby improving the performance and efficiency of Earth observation applications. In recent years, a growing body of research has focused on constructing such foundation models tailored for remote sensing. Mainstream

---

[*]First three authors contributed equally.
[†]Corresponding author: `t.huang2@exeter.ac.uk`

39th Conference on Neural Information Processing Systems (NeurIPS 2025) Track on Datasets and Benchmarks.

models can be divided into unimodal pre-trained foundation models (e.g., SatMAE [5], RingMo [6], ScaleMAE [7], SpectralGPT [8]) and vision-language foundation models (e.g., RemoteCLIP [9], GeoRSCLIP [10], RSGPT [11], GeoChat [12]). These EOFMs have demonstrated their powerful performance in numerous downstream tasks. A comprehensive review of EOFMs can be found in [13–19].

Despite advancements in EOFMs, there remains a significant gap in systematically benchmarking their robustness towards image perturbations. Remote sensing images are particularly susceptible to factors such as weather conditions or sensor discrepancies, which can introduce significant noise and variability [20–22], posing challenges to current EOFMs. Therefore, developing a comprehensive benchmark to evaluate and compare the robustness of these models holds great academic and practical significance. Such benchmarking efforts can guide the design of highly robust EOFMs that effectively adapt to noise and data variability, ensuring stable and reliable results under diverse conditions.

To achieve this, we introduce **REOBench**, a comprehensive **Bench**mark designed to evaluate the **R**obustness of **E**arth **O**bservation foundation models, covering state-of-the-art models based on masked image modeling, contrastive learning, and large language models. REOBench focuses on high-resolution optical remote sensing images, which are widely used in real-world applications such as urban planning and disaster response. We conducted experiments on **six** widely studied remote sensing image understanding tasks, covering both vision-centric and vision-language tasks, under **twelve** types of perturbations. These include both appearance-based corruptions (e.g., noise, blur, haze) and geometric distortions (e.g., rotation, scale, translation), applied at varying severity levels to simulate realistic environmental and sensor-induced challenges. Our evaluation yields three key findings:

⋆ Existing Earth observation foundation models suffer noticeable performance degradation under common image corruptions, with particularly sharp drops for the models based on masked image modeling.

⋆ The degree of vulnerability to image corruptions varies across tasks, model architectures, and types of perturbations, with performance drop varying from less than 1% to over 20%.

⋆ Vision-language foundation models exhibit greater robustness to visual perturbations compared to vision-centric foundation models, particularly in image-level scene classification tasks.

In summary, REOBench provides the first large-scale, task-diverse, and perturbation-rich benchmark for evaluating robustness in EOFMs. It offers actionable insights for the research community and serves as a stepping stone toward building more reliable, generalizable, and trustworthy AI systems for Earth observation.

## 2 REOBench Dataset

To systematically evaluate the robustness of EOFMs, we construct a benchmark dataset by incorporating widely used remote sensing datasets spanning diverse tasks. Specifically, we include AID [23] for scene classification, ISPRS Potsdam [24] for semantic segmentation, DIOR [25] for object detection, and three subsets from VRSBench [26] for image captioning, visual question answering (VQA), and visual grounding. These datasets are selected based on their popularity, diversity of content, and relevance to the tasks under evaluation.

### 2.1 Corruptions in Remote Sensing Images

Remote sensing platforms are subject to a wide range of visual degradations that differ significantly from those encountered by ground-based cameras. To systematically evaluate the robustness of RSFMs, we construct a benchmark comprising **12 synthetic corruptions**, categorized into three types: *environmental*, *sensor-induced*, and *geometric*. Each corruption is generated using physically or statistically grounded procedures to ensure the resulting images remain photorealistic while faithfully reflecting failure modes commonly observed in satellite and UAV imagery.

**Environmental Corruptions.** Atmospheric and illumination variations constitute predominant environmental degradations. For instance, *Cloud* occlusions substantially obscure optical remote sensing data, severely impacting scene interpretability [27, 28]. Variations in *Brightness* resulting from shifting sun angles affect radiometric stability and degrade feature matching and object recognition

performance [29, 30]. *Haze*, caused by aerosol scattering, significantly lowers image contrast and impairs detection and classification accuracy [31, 32]. Following established benchmarks [33], we simulate these environmental corruptions using physically motivated image augmentation techniques.

**Sensor-induced Corruptions.** Imperfections during sensor capture or data transmission introduce various degradations. *Gaussian Blur*, indicative of defocusing or modulation transfer function (MTF) degradation, compromises tie-point accuracy and feature localization [34, 35]. *Motion Blur*, arising from platform vibrations or rapid movements, negatively impacts object detection and tracking in aerial inspections [36]. *Gaussian Noise* and *Salt & Pepper Noise*, simulating electronic interference and bit-flip errors respectively, significantly decrease segmentation and classification accuracy [37, 38]. *Sensor Gap* degradations, exemplified by the Landsat-7 SLC-off issue, necessitate specialized gap-filling methodologies [39, 40]. Furthermore, *Compression* artifacts, such as those from JPEG/JPEG2000, substantially impair the quality of CNN feature extraction [41]. These sensor-induced corruptions are replicated through established augmentation and simulation protocols in line with existing research [33, 42].

**Geometric Corruptions.** Geometric distortions originate primarily from variations in sensor orientation, altitude, and registration accuracy. *Rotation* caused by platform roll or yaw introduces inconsistencies in orientation-sensitive feature extraction processes [43]. *Scale* alterations resulting from altitude fluctuations pose significant challenges for detectors lacking robust multi-scale adaptability [44]. *Translation*, modeling inaccuracies due to GPS drift, registration errors, or parallax, adversely affects pixel-aligned or patch-based analysis methods [45]. To effectively simulate these geometric degradations, we apply spatial transformations, including image *Rotation*, *Scaling*, and *Translation*, to remote sensing images.

In total, these corruption categories encompass twelve distinct types. Each type of corruption is applied consistently across all datasets at five severity levels. Fig. 1 illustrates one example of original and corrupted images.

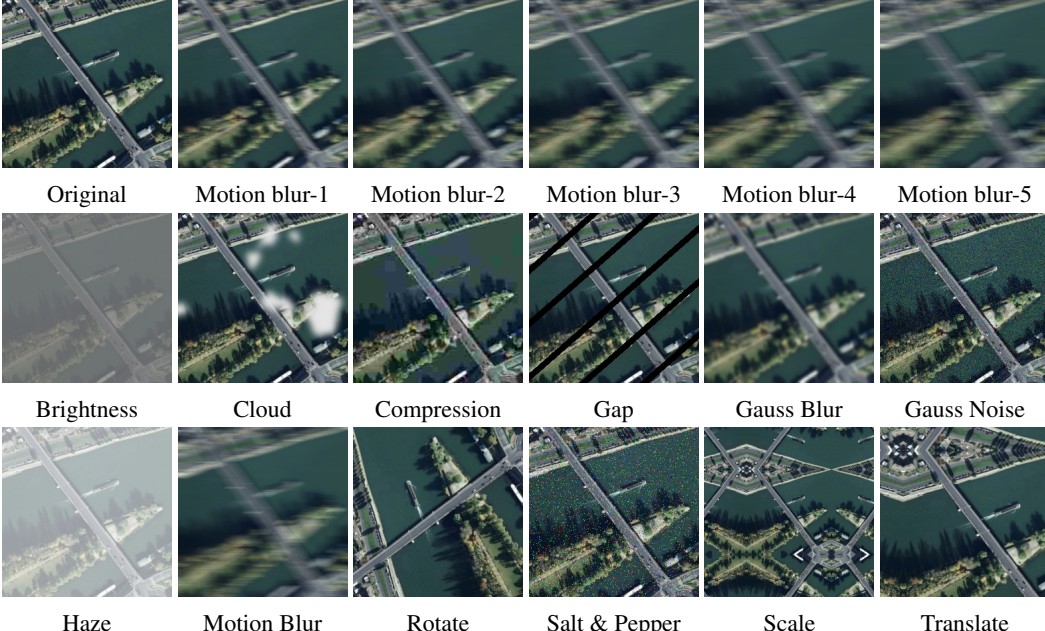

Figure 1: Example of perturbed images. In the first row, we present the original clean image alongside images perturbed by five levels of motion blur. The second and third rows illustrate examples of images corrupted by a range of perturbation types.

## 2.2 Definition of Corruption Robustness

We formalize the corruption robustness of EOFMs as its ability to maintain task performance in the presence of realistic geophysical and sensor-induced degradations frequently encountered in Earth

observation. Let $f : \mathcal{X} \to \mathcal{Y}$ denote an EOFMs that maps an input image $x \in \mathcal{X}$ to a label $y \in \mathcal{Y}$, with $(x, y)$ sampled from an underlying geospatial data-generating distribution $\mathcal{D}$. We define a set of corruptions $\mathcal{C} = \{c_1, \ldots, c_K\}$, where each $c_k \in \mathcal{C}$ represents a physically plausible corruption operator, such as haze, cloud occlusion, or sensor noise. Each corruption occurs with a non-zero prevalence $\mathbb{P}_{\mathcal{C}}(c_k) > 0$. To quantify robustness, we define the *relative task performance drop* ($\mathcal{R}_{\mathrm{TP}}$), which measures the degradation in model performance under corrupted inputs relative to its accuracy on clean data:

$$\mathcal{R}_{\mathrm{TP}} = \frac{\mathbb{P}_{(x,y)\sim\mathcal{D}}[f(x) = y] - \mathbb{E}_{c\sim\mathcal{C}}\left[\mathbb{P}_{(x,y)\sim\mathcal{D}}[f(c(x)) = y]\right]}{\mathbb{P}_{(x,y)\sim\mathcal{D}}[f(x) = y]}. \tag{1}$$

A *smaller* $\mathcal{R}_{\mathrm{TP}}$ indicates greater robustness, as it reflects less relative degradation when encountering corrupted data.

# 3 Benchmark Robustness on EOFMs

We evaluate the robustness of EOFMs across six widely studied remote sensing image understanding tasks: scene classification, semantic segmentation, object detection, image captioning, visual question answering (VQA), and visual grounding. The evaluated models represent the current state-of-the-art in remote sensing and can be broadly categorized into the following three types.

**MIM-based foundation models.** Masked image modeling (MIM) has gained popularity through the pioneering work of MAE [1]. In the field of remote sensing, notable approaches include SatMAE [5], RVSA [46], ScaleMAE [7], and SatMAE++ [47].

**CL-based foundation models.** Building on the success of the pioneering work CLIP [2], multiple contrastive learning (CL) -based foundation models have been introduced in the field of remote sensing, such as RemoteCLIP [9] and GeoRSCLIP [10]. To investigate robustness with respect to different backbone sizes, we evaluate two commonly used architectures in our experiments: ViT-B/32 and ViT-L/14 [48].

**LLM-based foundation models**. Following the pioneering works of GPT-4 [49], MiniGPT-4 [3], and LLaVA [4], multimodal large language models (MLLMs) have attracted significant research attention in recent years. Notable approaches include GeoChat [12], LHRS-Bot [50], RS-LLaVA [51], VHM [52], SkySenseGPT [53], and Falcon [54]. In our experiments, we evaluate models with open-access code and pretrained weights for comparison.

## 3.1 Implementation Details

For MIM- and CLIP-based models, we take the vision backbones from pretrained foundation models and append a task-specific head (e.g., MLP, detectors, or segmentors) for each task. For these LLM-based models, since these generalist models usually freeze their vision backbones and can naturally handle multiple tasks, we directly evaluate *zero-shot* performance of these models to test their robustness.

For the scene classification task, we take the backbone from all pretrained foundation models and append a single linear layer after the backbone for classification. For the semantic segmentation (resp. detection) task, we follow RVSA [46] to use UpperNet [55] (resp. Oriented R-CNN [56]) as the segmentor (resp. detector) and replace its backbone with that from pretrained foundation models. Following RVSA [46], we build a feature pyramid from blocks 4, 6, 8, and 12 using up/down-sampling. All models are trained for 12 epochs with an initial learning rate of 1e-5, decayed by 0.1 at epochs 8 and 11.

For vision-language tasks, including image captioning, visual question answering, and visual grounding, we follow the original paper designs to craft task-specific prompts and evaluate the zero-shot performance of these foundation models to assess robustness. It should be noted that different pretrained foundation models are designed to accept images at specific resolutions. When the input image size differs from the pretrained backbone's expected resolution, we interpolate the position embeddings in the backbone to accommodate the new input dimensions.

## 3.2 Scene Classification

From Table 1, we can draw the following findings: 1) All benchmark methods suffer from serious performance under image corruptions for the scene classification tasks, especially for MIM-based methods. 2) CL- and LLM-based methods are more robust towards image corruptions than MIM-based methods. This is probably because CL- and LLM-based methods are trained by matching image-text pairs in a shared embedding space, learning high-level semantic features less sensitive to low-level corruptions. In contrast, MIM-based methods are trained by reconstructing pixel- or token-level details, making them sensitive to local corruptions. Specifically, VHM [52] achieves the least performance drop under image corruptions. 3) CL-based methods usually perform better than MIM and LLM-based methods on the scene classification task, for both clean and noisy images. Specifically, GeoRSCLIP [10] achieves the best scene classification performance under image corruptions.

Table 1: Scene classification performance on AID dataset across different image perturbations. $zs$ denotes zero-shot evaluation.

| Method | Backbone | Clean | Brightness Contrast | Cloud | Compression Artifacts | Data Gaps | Gauss Blur | Gauss Noise | Haze | Motion Blur | Rotate | Salt Pepper | Scale | Translate | Avg | $\mathcal{R}_{TP}$ |
|---|---|---|---|---|---|---|---|---|---|---|---|---|---|---|---|---|
| MIM-based | | | | | | | | | | | | | | | | |
| SATLAS [57] | Swin-B | 90.85 | 82.54 | 84.32 | 73.36 | 67.23 | 78.10 | 79.16 | 80.46 | 32.44 | 72.54 | 77.56 | 72.54 | 88.54 | 74.07 | 18.47 |
| SatMAE [5] | ViT-L | 72.05 | 44.82 | 59.58 | 67.26 | 46.49 | 71.33 | 71.25 | 28.31 | 63.85 | 69.15 | 70.45 | 59.74 | 66.12 | 59.86 | 16.92 |
| Scale-MAE [7] | ViT-L | 75.75 | 51.80 | 72.65 | 39.60 | 43.69 | 31.65 | 46.31 | 55.24 | 17.49 | 66.15 | 47.27 | 61.58 | 69.84 | 50.27 | 33.64 |
| RVSA [46] | ViT-B | 84.60 | 56.84 | 77.33 | 56.07 | 53.14 | 53.53 | 32.51 | 49.19 | 23.45 | 76.88 | 35.12 | 71.78 | 77.22 | 55.26 | 34.69 |
| SatMAE++ [47] | ViT-L | 91.35 | 64.62 | 82.64 | 62.69 | 60.70 | 48.23 | 76.98 | 62.56 | 29.43 | 85.49 | 73.22 | 75.79 | 87.61 | 67.50 | 26.11 |
| CL-based | | | | | | | | | | | | | | | | |
| RemoteCLIP$_{zs}$ [9] | ViT-L | 81.10 | 78.32 | 80.64 | 73.91 | 79.43 | 76.83 | 76.72 | 80.10 | 57.02 | 82.80 | 70.90 | 68.39 | 80.72 | 75.48 | 6.93 |
| RemoteCLIP [9] | ViT-B | 96.85 | 90.80 | 95.36 | 91.13 | 88.96 | 89.18 | 94.25 | 87.46 | 63.75 | 96.22 | 91.43 | 83.62 | 95.42 | 88.97 | 8.15 |
| RemoteCLIP [9] | ViT-L | 95.45 | 93.11 | 93.80 | 88.77 | 94.21 | 92.47 | 94.20 | 93.37 | **74.45** | 95.01 | 86.99 | 83.37 | 94.06 | 90.32 | 5.38 |
| GeoRSCLIP$_{zs}$ [10] | ViT-L | 66.05 | 62.41 | 65.47 | 60.45 | 64.41 | 62.03 | 62.32 | 62.24 | 44.20 | 65.52 | 58.88 | 52.59 | 64.25 | 60.40 | 8.55 |
| GeoRSCLIP [10] | ViT-B | 96.90 | 93.59 | 96.04 | 91.01 | 93.34 | 92.60 | 92.99 | 92.91 | 57.78 | 95.70 | 88.22 | 75.70 | 93.87 | 88.65 | 8.51 |
| GeoRSCLIP [10] | ViT-L | **97.40** | **96.27** | **96.45** | **92.28** | **95.62** | **96.09** | **95.68** | **96.00** | 71.03 | **97.20** | **92.75** | **77.16** | 95.15 | **91.80** | 5.74 |
| LLM-based | | | | | | | | | | | | | | | | |
| GeoChat [12] | ViT-L | 65.85 | 64.67 | 65.26 | 60.71 | 64.61 | 63.32 | 62.34 | 64.54 | 48.68 | 65.05 | 62.32 | 56.21 | 62.91 | 61.72 | 6.27 |
| LHRS-Bot [50] | ViT-L | 87.75 | 87.38 | 86.82 | 78.53 | 85.95 | 84.94 | 82.62 | 87.62 | 67.76 | 87.31 | 76.73 | 79.07 | 86.71 | 82.79 | 5.65 |
| RS-LLaVA [51] | ViT-L | 67.55 | 65.36 | 68.95 | 63.05 | 67.69 | 65.73 | 63.13 | 65.97 | 43.86 | 66.55 | 68.24 | 54.89 | 64.89 | 63.07 | 6.63 |
| SkySenseGPT [53] | ViT-L | 87.35 | 87.72 | 87.91 | 79.66 | 87.67 | 84.78 | 83.08 | 87.71 | 63.11 | 86.86 | 83.61 | 75.49 | 85.25 | 82.74 | 5.28 |
| VHM [52] | ViT-L | 80.60 | 79.81 | 80.67 | 76.10 | 80.82 | 78.81 | 76.92 | 79.55 | 59.74 | 80.47 | 75.43 | 72.57 | 79.73 | 76.72 | **4.81** |

## 3.3 Semantic Segmentation

The ISPRS Potsdam dataset provides both non-eroded and eroded labels, corresponding to annotations with and without object boundaries, respectively. In our experiments, we use the non-eroded labels for model evaluation and report the mean IoU for MIM- and CL-based methods. We omit LLM-based methods for semantic segmentation due to the lack of open-source MLLMs for this task. From Table 2, we can draw the following findings: 1) both MIM- and CL-based methods suffer from serious performance under image corruptions for the object detection task, with a mIoU drop of more than 10%. 2) MIM-based methods achieve much better performance than CL-based methods on clean and noisy images. This is probably because MIM-based methods can capture local details by pixel reconstruction, while CL-based methods align global visual embeddings with text features, thus losing fine-grained details. Specifically, ScaleMAE [7] achieves the best segmentation performance under image corruption, with an average mIoU of 60.02%. 3) MIM-based methods suffer from a more serious performance drop under image corruptions than CL-based methods. Specifically, GeoRSCLIP [10] achieves the best robustness across image corruptions, with a drop of 10.01% of mIoU under corruptions.

Table 2: Semantic segmentation performance (mIoU) on the ISPRS Potsdam dataset under different image perturbations.

| Method | Backbone | Clean | Brightness Contrast | Cloud | Compression Artifacts | Data Gaps | Gauss Blur | Gauss Noise | Haze | Motion Blur | Rotate | Salt Pepper | Scale | Translate | Avg | $\mathcal{R}_{TP}$ |
|---|---|---|---|---|---|---|---|---|---|---|---|---|---|---|---|---|
| MIM-based | | | | | | | | | | | | | | | | |
| SatMAE [5] | ViT-L | 59.51 | 50.18 | 37.39 | 48.23 | 51.15 | 57.89 | 41.83 | 44.95 | 57.52 | 56.02 | 36.07 | 54.81 | 59.09 | 49.59 | 16.67 |
| ScaleMAE [7] | ViT-L | 68.92 | 64.37 | 65.43 | **49.96** | 41.84 | 64.86 | **54.89** | 63.89 | 64.49 | 64.51 | **52.12** | 65.45 | 68.38 | **60.02** | 12.91 |
| RVSA [46] | ViT-B | **69.82** | **64.71** | **65.67** | 47.99 | 45.34 | **66.89** | 48.89 | 61.52 | **65.25** | **65.58** | 45.26 | **66.87** | **69.23** | 59.43 | 14.88 |
| SatMAE++ [47] | ViT-L | 62.68 | 53.91 | 59.06 | 49.74 | **53.94** | 60.38 | 44.32 | 48.80 | 60.44 | 58.34 | 39.45 | 58.13 | 61.94 | 54.04 | 13.78 |
| CL-based | | | | | | | | | | | | | | | | |
| RemoteCLIP [9] | ViT-B | 50.28 | 42.32 | 45.27 | 39.33 | 37.78 | 50.26 | 48.46 | 36.61 | 50.06 | 46.48 | 46.91 | 48.39 | 49.63 | 45.12 | 10.26 |
| RemoteCLIP [9] | ViT-L | 56.69 | 54.51 | 52.73 | 43.24 | 51.19 | 50.82 | 45.12 | 50.47 | 51.82 | 53.68 | 38.98 | 54.59 | 56.53 | 50.31 | 11.25 |
| GeoRSCLIP [10] | ViT-B | 51.44 | 42.89 | 46.41 | 40.37 | 38.64 | 51.35 | 49.79 | 38.56 | 51.15 | 47.89 | 48.24 | 49.28 | 50.87 | 46.29 | **10.01** |
| GeoRSCLIP [10] | ViT-L | 56.81 | 54.97 | 52.53 | 43.28 | 41.37 | 50.49 | 42.7 | 49.41 | 51.36 | 53.54 | 36.98 | 54.66 | 56.64 | 48.99 | 13.77 |

## 3.4 Object Detection

We evaluate robustness on the corrupted images from the DIOR dataset. We report mAP for MIM- and CL-based methods. We omit LLM-based methods for the object detection task due to the lack of open-source LLMs for this task. From Table 3, we can draw the following findings: 1) both MIM- and CL-based methods suffer from serious performance under image corruptions, with a mAP drop of more than 8%. 2) MIM- and CL-based methods achieve comparable performance for object detection on clean and noisy images. RVSA [46] attains the highest mAP of 70.96% on clean images but experiences the most severe performance decline under image corruptions. 3) MIM-based and CL-based methods exhibit a similar degree of performance degradation when subjected to image corruptions. Among them, SatMAE++ [47] demonstrates the most robust detection performance under noisy conditions.

Table 3: Object detection performance (mAP) on the DIOR dataset across different image perturbations.

| Method | Backbone | Clean | Brightness Contrast | Cloud | Compression Artifacts | Data Gaps | Gauss Blur | Gauss Noise | Haze | Motion Blur | Rotate | Salt Pepper | Scale | Translate | Avg | $\mathcal{R}_{TP}$ |
|---|---|---|---|---|---|---|---|---|---|---|---|---|---|---|---|---|
| | | | | | MIM-based | | | | | | | | | | | |
| SatMAE [5] | ViT-L | 62.30 | 56.84 | 57.86 | 55.80 | 58.36 | 55.38 | 58.44 | 59.34 | 56.92 | 56.60 | 53.76 | 51.58 | 60.90 | 56.82 | 8.81 |
| ScaleMAE [7] | ViT-L | 70.20 | 64.80 | 65.98 | 62.50 | 64.46 | 62.58 | **63.82** | 66.10 | **63.08** | 63.44 | **60.50** | 53.08 | 68.26 | 63.22 | 9.94 |
| RVSA [46] | ViT-B | **70.96** | 60.59 | 65.02 | 61.58 | 64.60 | 62.35 | 62.87 | 63.98 | 62.88 | **64.04** | 56.61 | 55.97 | **69.69** | 62.51 | 11.91 |
| SatMAE++ [47] | ViT-L | 65.20 | 59.44 | 61.02 | 60.30 | 59.88 | 59.66 | 61.06 | 61.72 | 59.56 | 59.14 | 58.64 | 48.48 | 64.70 | 59.47 | **8.79** |
| | | | | | CL-based | | | | | | | | | | | |
| RemoteCLIP [9] | ViT-B | 60.40 | 56.72 | 56.28 | 56.78 | 54.56 | 53.68 | 57.36 | 55.90 | 53.42 | 54.54 | 54.40 | 44.92 | 59.72 | 54.86 | 9.17 |
| RemoteCLIP [9] | ViT-L | 70.20 | **66.52** | **66.62** | 63.84 | **65.40** | 63.62 | 63.68 | **66.76** | 62.66 | 63.52 | 59.16 | **57.42** | 68.64 | 63.99 | 8.85 |
| GeoRSCLIP [10] | ViT-B | 60.20 | 56.28 | 56.04 | 56.08 | 55.46 | 53.38 | 56.92 | 55.50 | 53.38 | 53.98 | 53.48 | 46.98 | 59.32 | 54.73 | 9.09 |
| GeoRSCLIP [10] | ViT-L | 69.80 | 66.12 | 65.34 | **65.34** | 64.96 | **63.62** | 62.90 | 66.04 | 62.02 | 62.68 | 56.04 | 57.40 | 68.10 | **63.38** | 9.20 |

## 3.5 Image Captioning

Table 4 presents the zero-shot image captioning performance of GeoChat [12], SkySenseGPT [53], VHM [52], RS-LLaVA [51], and the recently introduced Falcon model [54]. Following the VRS-Bench protocol [26], caption quality is evaluated using the GPT-4-based CLAIR metric [58][3]. Given that geometric distortions—such as rotation, scaling, and translation—can substantially alter image content, we exclude performance measurements under these corruption conditions for image captioning, VQA, and visual grounding tasks.

As shown in the upper part of Table 4, all models experience performance degradation under noisy conditions. Among them, the Falcon [54] model achieves the best overall performance, significantly outperforming other methods on both clean and corrupted images. However, it also suffers the largest performance drop of 6.28%. In contrast, RS-LLaVA [51] demonstrates the strongest robustness to image corruptions, exhibiting the smallest decrease in CLAIR score, with only a 2.03% drop. Additionally, we present results for GeoChat [12], fine-tuned on the VRSBench training set, as shown in the lower part of Table 4. The fine-tuned GeoChat model on the target dataset exhibits significantly improved performance compared to its zero-shot counterpart, with similar performance drop under corruptions.

Table 4: Image captioning performance (CLAIR) on the VRSBench-Cap dataset across different image perturbations. *ft* denotes models trained on the VRSBench training set.

| Method | Backbone | Clean | Brightness Contrast | Cloud | Compression Artifacts | Data Gaps | Gauss Blur | Gauss Noise | Haze | Motion Blur | Salt Pepper | Avg | $\mathcal{R}_{TP}$ |
|---|---|---|---|---|---|---|---|---|---|---|---|---|---|
| GeoChat [12] | ViT-L | 41.39 | 40.06 | 40.45 | 37.65 | 40.20 | 39.76 | 38.48 | 40.38 | 39.92 | 37.61 | 39.59 | 4.35 |
| SkySenseGPT [53] | ViT-L | 48.29 | 47.21 | 46.64 | 44.22 | 46.25 | 45.52 | 44.97 | 46.14 | 45.13 | 44.36 | 45.60 | 5.57 |
| VHM [52] | ViT-L | 52.02 | 50.19 | 50.82 | 50.26 | 50.57 | 51.22 | 50.46 | 50.39 | 50.72 | 49.48 | 50.46 | 3.00 |
| RS-LLaVA [51] | ViT-L | 51.30 | 51.15 | 50.43 | 51.78 | 50.54 | 52.01 | 47.84 | 50.57 | 49.88 | 48.12 | 50.26 | **2.03** |
| Falcon [54] | DaViT-B | **61.90** | **59.98** | **60.09** | **57.13** | **59.48** | **57.43** | **56.31** | **59.85** | **59.94** | **51.83** | **58.01** | 6.28 |
| GeoChat$_{ft}$ [12] | ViT-L | 71.26 | 69.00 | 68.93 | 66.60 | 69.45 | 68.63 | 67.83 | 69.98 | 69.02 | 63.87 | 68.15 | 4.36 |

## 3.6 Visual Question Anaswering

Table 5 reports VQA performance across various image perturbations. Following the VRSBench protocol [26], VQA performance is evaluated using the GPT-4-based matching accuracy[4]. From

---

[3]We use the `gpt-4o-mini-2024-07-18` model to compute the CLAIR scores.

[4]We use the `gpt-4o-mini-2024-07-18` model to compute the matching accuracy for VQA.

Table 5, it is evident that all LLM-based models experience a moderate decline in performance under image perturbations. Overall, VHM [52] achieves the best accuracy across both clean and noisy images. LHRS-Bot [50], RS-LLaVA [51], and Falcon [54], despite showing relatively lower overall accuracy, exhibit less sensitivity to image corruptions. Additionally, the GeoChat [12] fine-tuned on the VRSBench training set surpasses zero-shot models in terms of absolute performance, while exhibiting a slightly smaller performance drop under perturbations, indicating improved robustness.

Table 5: VQA performance (Accuracy) on the VRSBench-VQA dataset across different image perturbations. *ft* indicates models fine-tuned on the VRSBench training set.

| Method | Backbone | Clean | Brightness Contrast | Cloud | Compression Artifacts | Data Gaps | Gauss Blur | Gauss Noise | Haze | Motion Blur | Salt Pepper | Avg | $\mathcal{R}_{TP}$ |
|--------|----------|-------|--------------------|-------|----------------------|-----------|-----------|------------|------|------------|------------|-----|-----|
| GeoChat [12] | ViT-L | 56.63 | 53.89 | 54.82 | 55.14 | 55.99 | 55.88 | 55.44 | 56.22 | 54.08 | 54.04 | 55.06 | 2.77 |
| LHRS-Bot [50] | ViT-L | 35.72 | 35.72 | 35.69 | 35.72 | 35.72 | 35.72 | 35.72 | 35.72 | 35.34 | 35.72 | 35.56 | **0.45** |
| SkySenseGPT [53] | ViT-L | 60.21 | 59.26 | 59.73 | 57.93 | 59.64 | 59.21 | 58.27 | 59.63 | 59.17 | 57.27 | 58.90 | 2.18 |
| VHM [52] | ViT-L | **61.72** | **60.91** | **61.07** | **60.40** | **61.49** | **60.91** | **60.91** | **61.12** | **59.97** | **60.39** | **60.90** | 1.33 |
| RS-LLaVA [51] | ViT-L | 57.25 | 57.04 | 57.14 | 55.45 | 57.25 | 57.14 | 55.97 | 57.21 | 55.25 | 55.82 | 56.47 | 1.36 |
| Falcon [54] | DaViT-B | 33.27 | 32.83 | 32.70 | 32.19 | 33.30 | 33.43 | 32.85 | 32.76 | 32.97 | 31.55 | 32.73 | 1.59 |
| GeoChat$_{ft}$ [12] | ViT-L | 75.79 | 75.13 | 74.97 | 73.84 | 75.63 | 74.89 | 74.46 | 75.43 | 74.76 | 72.77 | 74.65 | 1.50 |

## 3.7 Visual Grounding

Table 6 presents the zero-shot visual grounding performance of comparing methods. We report grounding accuracy at an IoU threshold of 0.5. As shown in the upper part of Table 6, all methods experience noticeable declines in performance under image perturbations. The GeoGround [59] model achieves the best performance on both clean and perturbed images, with a grounding accuracy of 75.93% and the smallest drop of 4.48%. In comparison, the Falcon [54] model, despite not being trained on VRSBench, demonstrates competitive visual grounding capability, but experiences a more significant degradation in performance when exposed to image corruptions. The fine-tuned GeoChat [12] model shows substantial improvements over its zero-shot counterpart, with a substantially reduced performance drop under noisy conditions.

Table 6: Visual grounding performance on the VRSBench-Ref dataset across different image perturbations. We report grounding accuracy at an IoU threshold of 0.5. * indicates the GeoGround model includes VRSBench in its training data. *ft* indicates models fine-tuned on the VRSBench training set.

| Method | Backbone | Clean | Brightness Contrast | Cloud | Compression Artifacts | Data Gaps | Gauss Blur | Gauss Noise | Haze | Motion Blur | Salt Pepper | Avg | $\mathcal{R}_{TP}$ |
|--------|----------|-------|--------------------|-------|----------------------|-----------|-----------|------------|------|------------|------------|-----|-----|
| GeoChat [12] | ViT-L | 18.96 | 17.09 | 16.54 | 16.52 | 16.19 | 16.61 | 16.93 | 17.09 | 16.91 | 16.57 | 16.72 | 11.81 |
| VHM [52] | ViT-L | 37.20 | 34.66 | 35.29 | 34.18 | 35.48 | 35.01 | 35.78 | 35.54 | 32.21 | 34.58 | 34.74 | 6.61 |
| GeoGround* [59] | ViT-L | **75.93** | **73.57** | **71.57** | **71.30** | **72.23** | **73.23** | **72.92** | **74.06** | **72.11** | **71.77** | **72.53** | **4.48** |
| Falcon [54] | DaViT-B | 73.30 | 71.31 | 69.92 | 65.83 | 68.61 | 70.79 | 64.28 | 71.04 | 68.17 | 59.53 | 67.72 | 7.61 |
| GeoChat$_{ft}$ [12] | ViT-L | 55.50 | 53.79 | 52.20 | 50.51 | 53.06 | 53.11 | 51.57 | 54.23 | 52.99 | 49.82 | 52.36 | 5.66 |

## 4 Discussion

In this section, we further analyze the robustness of EOFMs across model architectures, tasks, corruption categories, and backbone sizes.

### 4.1 Vision-Centric vs. Vision-Language Foundation Models

As shown in Fig. 2, vision-centric foundation models (MIM-based) tend to suffer greater performance degradation under visual perturbations compared to vision-language models (CL- and VLM-based). This difference is especially pronounced in image-level scene classification tasks, where MIM-based models exhibit an average performance drop exceeding 25%. In contrast, vision-language models consistently demonstrate stronger robustness across tasks, maintaining performance drops below 10% in most cases. This is probably due to the complementary grounding effect of language supervision. We also note that the robustness gap between vision-centric and vision-language models is less significant for segmentation and detection tasks.

## 4.2 Robustness Across Different Tasks

Fig. 2 further highlights that vulnerability to perturbations varies substantially across tasks. MIM-based models are particularly sensitive in classification tasks, while CL-based models maintain greater stability across classification, segmentation, and detection tasks. This can be attributed to the contrastive objective, which encourages learning of invariant and robust representations. LLM-based models, on the other hand, show the smallest performance degradation in vision-language tasks such as image captioning and visual question answering (VQA)—typically below 5%. These results suggest that LLM-based methods excel in corruption-robust generalization, particularly in tasks that benefit from multimodal alignment.

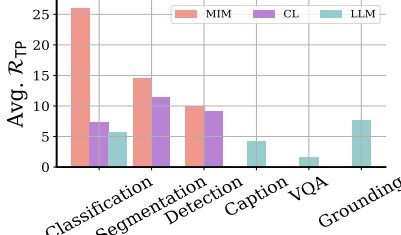

Figure 2: Robustness across different tasks and model architectures. We report the average $\mathcal{R}_{\text{TP}}$ across models.

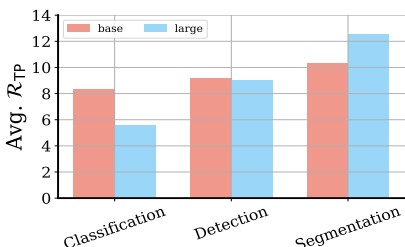

Figure 3: Robustness across different backbone sizes. We report the average $\mathcal{R}_{\text{TP}}$ for RemoteCLIP and GeoRSCLIP.

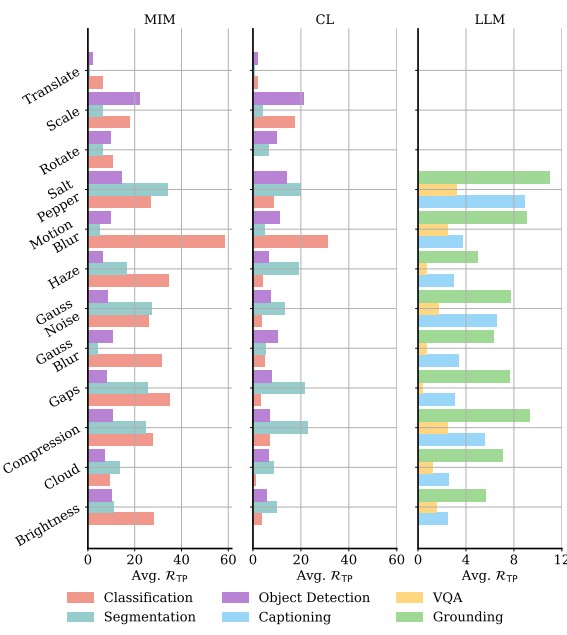

Figure 4: Robustness across different types of corruptions. We report the $\mathcal{R}_{\text{TP}}$ across models.

## 4.3 Robustness Across Different Backbone Sizes

For scene classification, the larger backbone (ViT-L) demonstrates greater robustness, showing less performance degradation under image corruptions. However, for fine-grained tasks such as semantic segmentation and object detection, ViT-L suffers larger performance drops compared to ViT-B. This suggests that while a larger backbone may enhance robustness in high-level recognition tasks, it may also amplify sensitivity to image corruptions in pixel- or region-level tasks.

## 4.4 Robustness Across Perturbation Types

As shown in Fig. 4, the performance degradation of MIM-, CL-, and LLM-based models varies notably across different visual perturbations. Motion blur causes the most severe drop, especially for MIM, which loses around 60% in performance, indicating a high sensitivity to spatial distortions. In contrast, translation has the least impact, suggesting minimal disruption to pattern recognition. Across perturbation types, LLM-based models consistently exhibit the strongest robustness, maintaining performance drops below 10% in most cases. This reinforces the value of language supervision in promoting the learning of more semantic and perturbation-invariant features.

## 4.5 Robustness Across Compound Perturbations

Real-world images often suffer from multiple simultaneous degradations (e.g., haze combined with noise). To investigate this, we evaluate object detection performance under selected compound perturbations on the DIOR-R dataset. The results from Table 7 reveal that models exhibit substantially lower accuracy under compound perturbations compared to single perturbations. For example, the detection performance of RemoteCLIP drops from 56.72% under Brightness to 40.58% under all three combined perturbations. Moreover, certain combinations of perturbations, such as *Brightness + Compression*, produce synergistic effects, causing performance declines that exceed the sum of individual perturbation effects.

Table 7: Object Detection performance (mAP) on DIOR-R under compound corruptions

| Model | Clean | Brightness | Clouds | Compression | Brightness + Clouds | Brightness + Compression | Clouds + Compression | All Three |
|---|---|---|---|---|---|---|---|---|
| Brightness | ✗ | ✓ | ✗ | ✗ | ✓ | ✓ | ✗ | ✓ |
| Clouds | ✗ | ✗ | ✓ | ✗ | ✓ | ✗ | ✓ | ✓ |
| Compression | ✗ | ✗ | ✗ | ✓ | ✗ | ✓ | ✓ | ✓ |
| RemoteCLIP | 60.40 | 56.72 | 56.28 | 56.78 | 49.98 | 45.36 | 52.57 | 40.58 |
| GeoRSCLIP | 60.20 | 56.28 | 56.04 | 56.08 | 50.27 | 44.90 | 52.39 | 40.94 |

## 4.6 Robustness in Multispectral Remote Sensing

In addition to high-resolution RGB imagery, multispectral data provide complementary spectral information that can improve scene understanding and robustness in Earth Observation. To assess the robustness of EO foundation models beyond optical imagery, we conduct preliminary experiments on two representative multispectral datasets: fMoW-Sentinel2 [60] and BigEarthNet [61], using two recently proposed foundation models, SatMAE [5] and EarthDial [62]. The results, summarized in Table 8, show substantial performance drops under image perturbations, underscoring the brittleness of current EO foundation models when applied to multispectral data.

Table 8: Scene classification performance on the multispectral dataset across different image perturbations.

| Method | Backbone | Clean | Brightness Contrast | Cloud | Compression Artifacts | Data Gaps | Gauss Blur | Gauss Noise | Haze | Motion Blur | Rotate | Salt Pepper | Scale | Translate | Avg | $\mathcal{R}_{TP}$ |
|---|---|---|---|---|---|---|---|---|---|---|---|---|---|---|---|---|
| SatMAE [5] | fMoW-S2 [60] | 59.75 | 37.46 | 58.69 | 33.58 | 50.01 | 29.35 | 36.64 | 41.57 | 40.12 | 38.93 | 51.79 | 43.35 | 59.68 | 43.43 | 27.31 |
| EarthDial [62] | BigEarthNet [61] | 46.521 | 31.47 | 46.48 | 45.40 | 34.97 | 36.37 | 38.69 | 33.25 | 39.84 | 29.51 | 22.71 | 39.18 | 45.30 | 36.93 | 20.62 |

# 5 Related Works

**Robustness Research in Remote Sensing.** Deep learning (DL)-based methods have achieved significant success in remote sensing image processing; however, their black-box nature raises concerns regarding interpretability, transparency, and vulnerability to adversarial examples. Recent studies have begun to address the robustness of DL models in this domain. Kazmi et al.[20] present a comprehensive literature review on adversarial attacks in aerial imagery processing, but do not provide an in-depth analysis of model robustness. Mei et al.[21] examine the robustness of DL-based methods for remote sensing image understanding, with a focus on image classification and object detection tasks. Lian et al. [63, 22] propose techniques to enhance adversarial robustness specifically for object detection in aerial imagery. [64] aims to improve the adversarial robustness of scene classification models in remote sensing via CAM-guided feature learning. These works only study the robustness of task-specific models. In contrast, our work for the first time investigates the robustness of foundation models in remote sensing.

**Foundation Models in Remote Sensing.** In general, there are four types of foundation models in remote sensing: MIM-based, CL-based, LLM-based, and diffusion-based methods. (1) Masked Image Modeling (MIM) has gained popularity through the pioneering work of MAE [1]. These methods typically employ an encoder network to learn feature representations by masking a portion of the image tokens, followed by a decoder network that reconstructs the masked image pixels in a self-supervised manner. In the field of remote sensing, notable approaches include SatMAE [5],

RingMo [6], RVSA [46], ScaleMAE [7], SatMAE++ [47], and DOFA [65]. (2) Contrastive Learning (CL) employs separate encoders to project images and texts into a shared embedding space, using a contrastive objective to align the resulting embeddings. Building on the success of the pioneering work CLIP [2], several contrastive learning-based foundation models have been introduced in the field of remote sensing, including RS-CLIP [10], RemoteCLIP [9], GeoRSCLIP [10], SkyCLIP [66], S-CLIP [67], SatCLIP [68], and GeoCLIP [69]. (3) Following the pioneering works of MiniGPT-4 [3] and LLaVA [4], multimodal large language models (MLLMs) have attracted significant research attention in recent years. For instance, RSGPT [11] introduces the first GPT-based MLLM tailored for remote sensing image understanding. Other notable approaches include GeoChat [12], EarthGPT [70], EarthMarker [71], Popeye [72], RS-LLaVA [51], VHM [52], LHRS-Bot [50], SkyEyeGPT [73], SkySenseGPT [53], RSUniVLM [74], and Falcon [54]. (4) Diffusion-based foundation models learn the joint distribution between text prompts and images through a forward noising process followed by a reverse denoising process. Recent studies have applied these models to synthesize satellite [75, 76], aerial [77], hyperspectral [78], and multi-resolution imagery [79].

## 6   Conclusion and Future Work

In this work, we present REOBench, the first comprehensive benchmark for evaluating the robustness of EOFMs across six core tasks and twelve perturbation types. Our evaluation reveals that existing EOFMs experience noticeable performance degradation under image corruptions. We also observe significant variations in robustness across model types, task categories, and backbone sizes, offering valuable insights for future development of robust models. We hope REOBench will serve as a standard benchmark to drive the creation of more robust and reliable models for Earth observation.

Despite its contributions, this work has several limitations. First, the evaluation is limited to high-resolution optical imagery, excluding other key modalities such as multispectral (e.g., Sentinel-2), hyperspectral, and SAR data. Second, the benchmark's dataset and task coverage are not exhaustive. While it includes widely used datasets (AID, Potsdam, DIOR, VRSBench), they may not fully reflect global variation in geography, resolution, or sensor types. Additionally, important tasks such as change detection, region captioning, and object counting are currently not included.

## 7   Broader Impact

REOBench aims to improve the reliability of Earth observation foundation models by systematically evaluating their robustness to real-world noise and perturbations. This is critical for high-stakes applications such as disaster response and environmental monitoring. By identifying vulnerability patterns across tasks and models, our benchmark can guide the development of future robust models.

## Acknowledgments and Disclosure of Funding

This work is supported by the EDF-KOIOS project and used the Dutch national e-infrastructure with the support of the SURF Cooperative using the funding of the projects EINF-12538, EINF-10925 and NWO-2023.027. We would like to express our deepest gratitude to the anonymous reviewers whose insightful comments and suggestions significantly improved the quality of this paper.

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
