# Supplementary of REOBench: Benchmarking Robustness of Earth Observation Foundation Models

**Xiang Li**[1*], **Yong Tao**[2*], **Siyuan Zhang**[3*], **Siwei Liu**[4], **Zhitong Xiong**[5]
**Chunbo Luo**[2], **Lu Liu**[2], **Mykola Pechenizkiy**[6], **Xiao Xiang Zhu**[5], **Tianjin Huang**[2,6†]

[1] University of Bristol, UK [2] University of Exeter, UK
[3] South China Normal University, China [4] The University of Aberdeen, UK
[5] Technical University of Munich, Germany [6] Eindhoven University of Technology, NL

## 1 REOBench Documentation and Intended Uses

### 1.1 Overview

REOBench dataset includes four subsets: AID for scene classification, ISPRS Potsdam for semantic segmentation, DIOR for object detection, and VRSBench for image captioning, visual question answering (VQA), and visual grounding.

### 1.2 Data Organization

Our REOBench dataset is organized as follows.

```
root/
├── AID
│   ├── AID_train.zip
│   ├── AID_test.zip
│   └── AID_JSON
├── Potsdam
│   ├── Potsdam_Images_trian.zip
│   ├── Potsdam_Anns_trian.zip
│   ├── Potsdam_Images_test.zip
│   └── Potsdam_Anns_test.zip
├── DIOR
│   ├── DIOR_Images_trian.zip
│   ├── DIOR_Anns_trian.zip
│   ├── DIOR_Images_test
│   │   └── clean.zip; brightness_contrast.zip; cloud.zip;
│   │       compression_artifacts.zip...
│   └── DIOR_Anns_test
│       └── clean.zip; rotate.zip; scale.zip; translate.zip
└── VRSBench
    ├── VRSBench_Images_trian.zip
    ├── VRSBench_train.json
    ├── VRSBench_Images_test
    │   └── clean.zip; brightness_contrast.zip; cloud.zip;
    │       compression_artifacts.zip...
    └── VRSBench_EVAL_Cap.json
```

---

*First three authors contributed equally.

†Corresponding author: `t.huang2@exeter.ac.uk`

39th Conference on Neural Information Processing Systems (NeurIPS 2025) Track on Datasets and Benchmarks.

```
├── VRSBench_EVAL_referring.json
└── VRSBench_EVAL_vqa.json
```

Detailed descriptions for each folder or file are given below.

- AID/Images_train.zip contains all AID images in the training set.
- AID/Images_test.zip contains images in the test set under corruption.
- AID/AID_JSON folder contains json file for zero-shot evaluation of LLM-based models.
- Potsdam/Potsdam_Images_trian.zip contains all Potsdam images in the training set.
- Potsdam/Potsdam_Images_trian.zip contains all Potsdam images in the test set under corruption.
- Potsdam/Potsdam_Anns_trian.zip contains annotations for images in the training set.
- Potsdam/Potsdam_Anns_test.zip contains annotations for images in the test set under corruption.
- DIOR/DIOR_Images_trian.zip contains all DIOR images in the training set.
- DIOR/DIOR_Anns_trian.zip contains all Oriented Bounding Boxes annotations for images in the training set.
- DIOR/DIOR_Images_test folder contains all DIOR images in the test set under corruption.
- DIOR/DIOR_Anns_test folder contains oriented bounding box annotations for test images under four settings: clean, and three spatial transformations — rotate, scale, and translate. For corruptions that do not involve spatial transformations (e.g., blur, noise), annotations from the clean setting are reused, as these corruptions do not alter object positions or shapes.
- VRSBench/Annotation_Images_train.zip contains VRSBench images in the training set.
- VRSBench/Annotation_Images_test folder contains VRSBench images in the test set, one folder per noise type.
- VRSBench/VRSBench_train.json contains VRSBench training annotations following LLaVA in standard JSON format.
- VRSBench/VRSBench_EVAL_Cap.json contains VRSBench evaluation annotations for the captioning task in standard JSON format.
- VRSBench/VRSBench_EVAL_referring.json contains VRSBench evaluation annotations for the visual grounding task in standard JSON format.
- VRSBench/VRSBench_EVAL_vqa.json contains VRSBench evaluation annotations for the VQA task in standard JSON format.

## 1.3 Intended Uses

REOBench is intended for use in academic and research settings, specifically for:

- Evaluating the robustness of remote sensing foundation models.
- Understand the robustness of remote sensing foundation models across tasks, noise types, and model architectures.

## 1.4 Use Cases

- **Robustness Research**: REOBench provides a standardized testbed for evaluating the robustness of remote sensing foundation models under a wide range of realistic corruptions, making it valuable for both academic and applied robustness research.
- **Model Diagnosis and Comparison**: The benchmark enables fine-grained analysis of performance degradation across different model types (e.g., MIM-based, CL-based, LLM-based), tasks, and corruption types, serving as a practical tool for diagnosing model vulnerabilities and comparing architectures.
- **Guidance for Model Development**: Insights derived from REOBench can inform the design of more resilient remote sensing models and training strategies, particularly for mission-critical applications such as disaster response and environmental monitoring.

## 1.5 Limitations

- **Modal and Sensor Scope**: This benchmark focuses exclusively on optical remote sensing imagery and does not currently include multispectral, hyperspectral, or SAR modalities, limiting its applicability to other sensing systems.

- **Task Coverage**: While REOBench includes six core tasks, it does not encompass important tasks such as change detection, region captioning, and object counting, which are increasingly relevant in Earth observation.

- **Dataset Representativeness**: The benchmark is constructed from well-known datasets such as AID, DIOR, Potsdam, and VRSBench, which may not fully capture the geographic, temporal, and sensor diversity of real-world global remote sensing data.

## 1.6 Ethical Considerations

- **Public Data Usage**: REOBench is built entirely from publicly available datasets. No personal, private, or sensitive information is included, and care has been taken to respect privacy in all sourced imagery.

- **Responsible Use**: We encourage the research community to use this benchmark ethically, particularly in downstream applications involving environmental monitoring, urban analytics, and decision-making systems that may influence public policy or resource allocation.

## 1.7 Documentation and Maintenance

- **Versioning**: Detailed version history of the dataset will be maintained to track changes and improvements over time.

- **Version Control**: The benchmark will be versioned to ensure transparency in updates, including bug fixes, additional corruption types, and task expansions.

- **Community Feedback**: We welcome contributions and suggestions from the community to expand, refine, and validate the benchmark, fostering collaborative progress in building robust AI systems for Earth observation.

## 1.8 Accountability Framework

To promote responsible AI development, REOBench adopts an open accountability model. Users are encouraged to report any issues related to dataset quality, annotation errors, or robustness evaluation inconsistencies. These reports will be reviewed and integrated into future updates, supporting a continuous feedback loop for improving the benchmark's accuracy, fairness, and utility.

# 2 URL to Data and Metadata

The REOBench dataset can be accessed and downloaded through our Huggingface repository (`https://huggingface.co/datasets/xiang709/REOBench`). Detailed metadata for the dataset is documented using the Croissant metadata framework, ensuring comprehensive coverage and compliance with the MLCommons Croissant standards, check [metadata](`https://huggingface.co/api/datasets/xiang709/REOBench`).

# 3 Author Statement and Data License

**Author Responsibility Statement:** The authors bear all responsibilities in case of any violations of rights or ethical concerns regarding the REOBench dataset.

**Data License Confirmation:** The dataset is released under the [CC-BY-4.0], which permits unrestricted use, distribution, and reproduction in any medium, provided the original work is properly cited.

# 4 Hosting and Accessibility

The REOBench dataset is hosted on Huggingface (`https://huggingface.co/datasets/xiang709/REOBench`) to ensure reliable and continuous accessibility.

**Maintenance Plan:** Ongoing maintenance and updates will be managed by the dataset authors, with updates scheduled bi-annually or as significant changes in the data sources occur.

**Long-term Preservation:** The dataset is archived in Huggingface (`https://huggingface.co/datasets/xiang709/REOBench`) to ensure long-term availability.

**Structured Metadata:** The annotation for each image is well-organized in standard JSON format to ensure easy usage.

# 5 Dataset Collection Details

## 5.1 Source datasets

Our REOBench uses three source datasets, i.e., AID for scene classification, ISPRS Potsdam for semantic segmentation, DIOR for object detection, and VRSBench for image captioning, visual question answering (VQA), and visual grounding. The details of each dataset are shown in Table 1.

Table 1: Statistics of source datasets.

| Dataset | Train | Test | Category | Size |
|---------|-------|------|----------|------|
| AID | 8,000 | 2,000 | 30 | $256 \times 256$ |
| Potsdam | 3,456 | 2,016 | 6 | $512 \times 512$ |
| DIOR-R | 11,725 | 11,738 | 20 | $800 \times 800$ |
| VRSBench | 20,264 | 9,350 | 26 | $512 \times 512$ |

## 5.2 Image Pertubations

Table 2: Configurations of different image corruption types and corresponding severity.

| Corruption Type | Parameter | S1 | S2 | S3 | S4 | S5 |
|-----------------|-----------|-----|-----|-----|-----|-----|
| Gaussian Noise | $\sigma$ | 0.04 | 0.05 | 0.06 | 0.07 | 0.08 |
| Salt Pepper Noise | amount | 0.005 | 0.01 | 0.02 | 0.03 | 0.05 |
| Gaussian Blur | kernel size | 3×3 | 5×5 | 7×7 | 9×9 | 11×11 |
| Motion Blur | kernel size | 2×2 | 4×4 | 6×6 | 8×8 | 10×10 |
| Brightness/Contrast | b / c | +0.0 / 1.0 | +0.1 / 0.8 | +0.2 / 0.6 | +0.3 / 0.4 | +0.4 / 0.2 |
| Clouds | threshold | 0.90 | 0.85 | 0.80 | 0.75 | 0.70 |
| Haze | intensity | 0.20 | 0.30 | 0.40 | 0.50 | 0.60 |
| Data Gaps | num / width (px) | 2 / 3 | 3 / 4 | 4 / 5 | 5 / 6 | 6 / 7 |
| Compression Artifacts | JPEG quality | 30 | 25 | 20 | 15 | 10 |
| Rotation | angle (°) | 30 | 45 | 60 | 75 | 90 |
| Scaling | scale ratio | 0.9 | 0.8 | 0.7 | 0.6 | 0.5 |
| Translation | displacement (px) | ±15 | ±20 | ±25 | ±30 | ±35 |

In table 2, we report parameter configures for different noise levels and types. S1 to S5 represent the severity levels of each corruption, with S1 being the mildest and S5 the most severe. The "Parameter" column lists the key control variables for each type of image corruption. For Gaussian noise, $\sigma$ represents the standard deviation that controls the amplitude of the Gaussian noise; for salt-and-pepper noise, the `amount` specifies the proportion of pixels randomly replaced with black or white; Gaussian blur and motion blur use `kernel size` to define the size of the convolution kernel; brightness and contrast corruption is controlled using $b/c$ to represent brightness offset and contrast scaling respectively; for the cloud corruption, `density` denotes the cloud coverage, which is controlled based on the Perlin noise threshold [1]; the parameter for haze simulation is `intensity`, a blending ratio that determines the degree of mixing with a white layer, and in subsequent visual fidelity experiments we extend the severity levels up to 9 with corresponding intensity values of 0.70, 0.80, 0.90, and 0.95 for S6–S9, respectively; data gaps are defined by `number/width of stripes`,

which specifies the number and width of missing regions; Compression Artifacts are governed by the `JPEG quality` parameter, with lower values indicating stronger compression artifacts. In terms of geometric transformations, rotation is defined by `angle`, scaling is represented by `scale ratio`, and translation is described by the maximum offset in `displacement`. These parameters allow for systematic control and severity grading of each corruption type.

### 5.3 Perturbation Fidelity Analysis

To assess the visual fidelity of our synthetic perturbations, we compute the Fréchet Inception Distance (FID) [2] between corrupted images and their corresponding clean counterparts. Specifically, we report FID scores for four benchmarks with different image resolutions: DIOR (800×800), VRSBench (512×512), Potsdam (512×512), and AID (256×256). For each dataset, FID was calculated across five severity levels (1–5) of perturbations, using the clean dataset as the reference distribution. Table 3 shows that FID values remain consistently low across severity levels, suggesting that the corrupted datasets are visually similar to real-world imagery and therefore suitable for robustness evaluation.

Table 3: FID scores between corrupted datasets and their corresponding clean datasets across different severity levels. Lower values indicate higher visual similarity.

| Severity | DIOR (800×800) | VRSBench (512×512) | Potsdam (512×512) | AID (256×256) |
|---|---|---|---|---|
| 1 | 2.36 | 3.84 | 6.06 | 14.07 |
| 2 | 5.46 | 7.67 | 10.35 | 23.60 |
| 3 | 9.67 | 11.69 | 17.06 | 32.37 |
| 4 | 14.21 | 14.76 | 24.86 | 39.31 |
| 5 | 19.81 | 20.06 | 37.89 | 49.03 |

To evaluate the visual fidelity of our synthetic weather perturbations relative to real-world corruptions, we conduct experiments on two benchmark datasets: SEN12MS-CR [3] for cloud perturbations and RRSHID [4] for haze perturbations. For each dataset, we apply our synthetic perturbations to the clean subsets and compute FID against the corresponding real corrupted subsets, which serve as references for natural weather effects. As shown in Table 4, the FID scores remain below 130 across all severity levels, indicating that the generated perturbations are visually similar to real-world conditions. For cloud perturbations on SEN12MS-CR, the FID increases consistently with severity, suggesting that lower levels more closely resemble natural cloud coverage, whereas higher levels introduce progressively stronger distortions. For haze perturbations on RRSHID, the severity range is extended to level 9. The FID decreases up to level 5 and rises thereafter, implying that moderate haze (around level 5) best matches the distribution of real-world haze samples.

Table 4: Comparative FID scores for synthetic perturbations against real-world weather datasets across different severity levels. Lower values indicate higher visual similarity.

| Dataset | 1 | 2 | 3 | 4 | 5 | 6 | 7 | 8 | 9 |
|---|---|---|---|---|---|---|---|---|---|
| SEN12MS-CR (clouds) [3] | 105.56 | 117.27 | 123.72 | 125.70 | 127.03 | – | – | – | – |
| RRSHID (haze) [4] | 107.46 | 100.79 | 90.79 | 80.52 | 73.23 | 73.83 | 88.37 | 128.83 | 190.30 |

## 6 More Experimental Results

### 6.1 More Implementation Details

Table 5 lists training configurations for different tasks. For the scene classification task, all experiments are conducted on a single NVIDIA RTX 4090 GPU with 24 GB of memory, using the AID dataset, which contains 10,000 high-resolution aerial images. The dataset is split into 8,000 training samples and 2,000 testing samples, with all images uniformly resized to $256 \times 256$ pixels. All models are trained for 100 epochs using the Adam [5] optimizer with an initial learning rate of 1e-3 and a batch size of 16. A CosineAnnealingWarmRestarts [6] scheduler is employed, with the initial cycle length set to 5 epochs.

For the semantic segmentation task, all experiments are conducted on a single NVIDIA A100 GPU with 40 GB of memory. The experiments are implemented using the MMSegmentation [7] framework, with UPerHead used as the primary decoder head and FCNHead as the auxiliary head. All training and evaluation are performed on the Potsdam dataset, where each original image ($6,000 \times 6,000$ pixels) is divided into 144 non-overlapping patches of $512 \times 512$ pixels for model training and inference. The resulting dataset comprises 5,472 image patches, split into 3,456 samples for training and 2,016 for testing. Training is performed for 12 epochs using the AdamW optimizer [8], with an initial learning rate of 6e-5, a batch size of 8, and a weight decay of 0.05.

For the object detection task, all experiments are conducted on a single NVIDIA A100 GPU with 80 GB of memory. The experiments are implemented using the MMRotate framework [9], with Oriented R-CNN employed as the detection head. All images in the DIOR-R dataset are resized to $800 \times 800$ pixels. The official trainval split is used for training, while the test split is used for both clean evaluation and as the basis for our noise-augmented test set. For fine-tuning, we extracted the last-layer feature map from each model's pretrained backbone and constructed a four-level feature pyramid via up/down-sampling, which is then fed into a newly initialized Oriented R-CNN head. The detector is trained for 12 epochs using the AdamW optimizer [8], with an initial learning rate of 1e-5, a batch size of 1, and a learning rate decay of 0.1 applied at epochs 8 and 11.

Table 5: Training configurations for different tasks.

|  | Scene Classification | Semantic Segmentation | Object Detection |
|---|---|---|---|
| Dataset | AID | Potsdam | DIOR-R |
| Decoder | Linear | UpperNet | Oriented R-CNN |
| Optimizer | Adam | AdamW | AdamW |
| Epochs | 100 | 12 | 12 |
| Lr | 1e-3 | 6e-5 | 1e-5 |
| Batch Size | 16 | 8 | 1 |

## 6.2 Evaluation Prompts

For image captioning evaluation, we use GPT-4o-mini to determine for each image whether the predicted captions match the ground truth, with the prompt: *You are trying to tell if a candidate set of captions is describing the same image as a reference set of captions. Candidate set: {candidate_statements}. Reference set: {target_statements}. On a precise scale from 0 to 100, how likely is it that the candidate set is describing the same image as the reference set?(JSON format, with a key "score", value between 0 and 100.*

For VQA evaluation, we use GPT-4o-mini to determine for each question whether the answers match ground truth texts, with the prompt: *Question: {question}, Ground Truth Answer: {ground_truth}, Predicted Answer: {predicted answer}. Does the predicted answer match the ground truth? Answer 1 for match and 0 for not match. Use semantic meaning not exact match. Synonyms are also treated as a match, e.g., pond and swimming pool.*

## 6.3 Qualitative Results

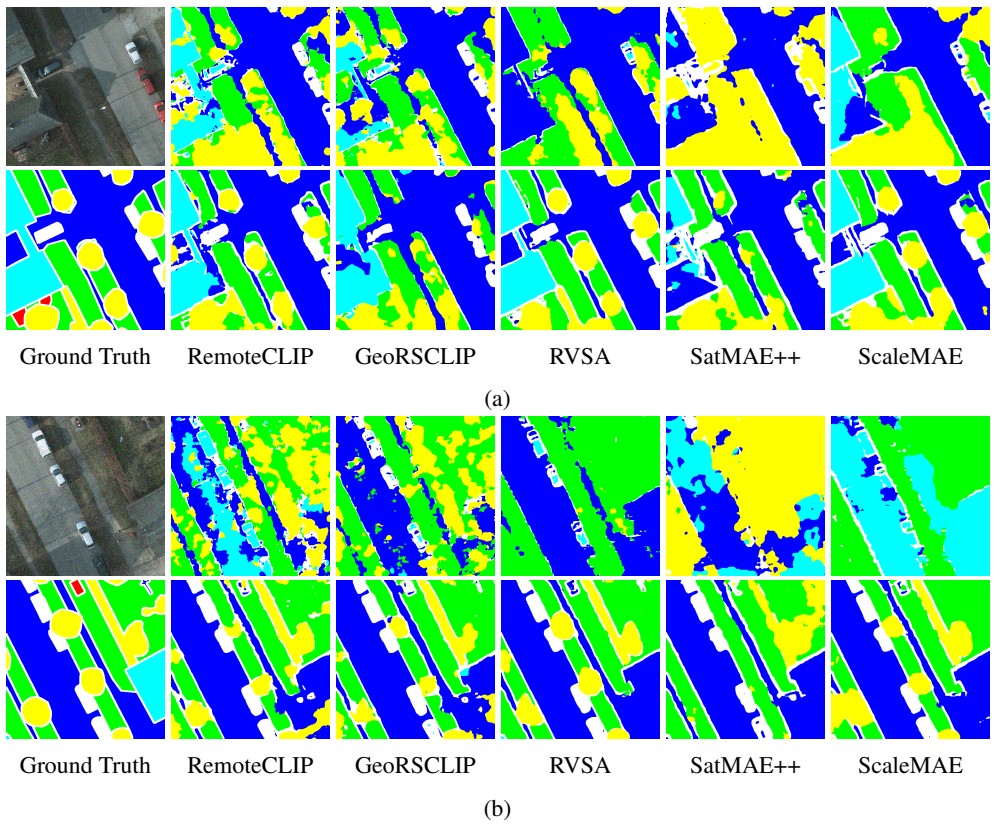

Figure 1: Selected semantic segmentation examples under (a) Gaussian noise (severity 5) and (b) Salt-and-pepper noise (severity 5). For each example, the top row shows the corrupted image and the segmentation results of different models under this corruption, and the bottom row shows the ground truth and segmentation results on clean images.

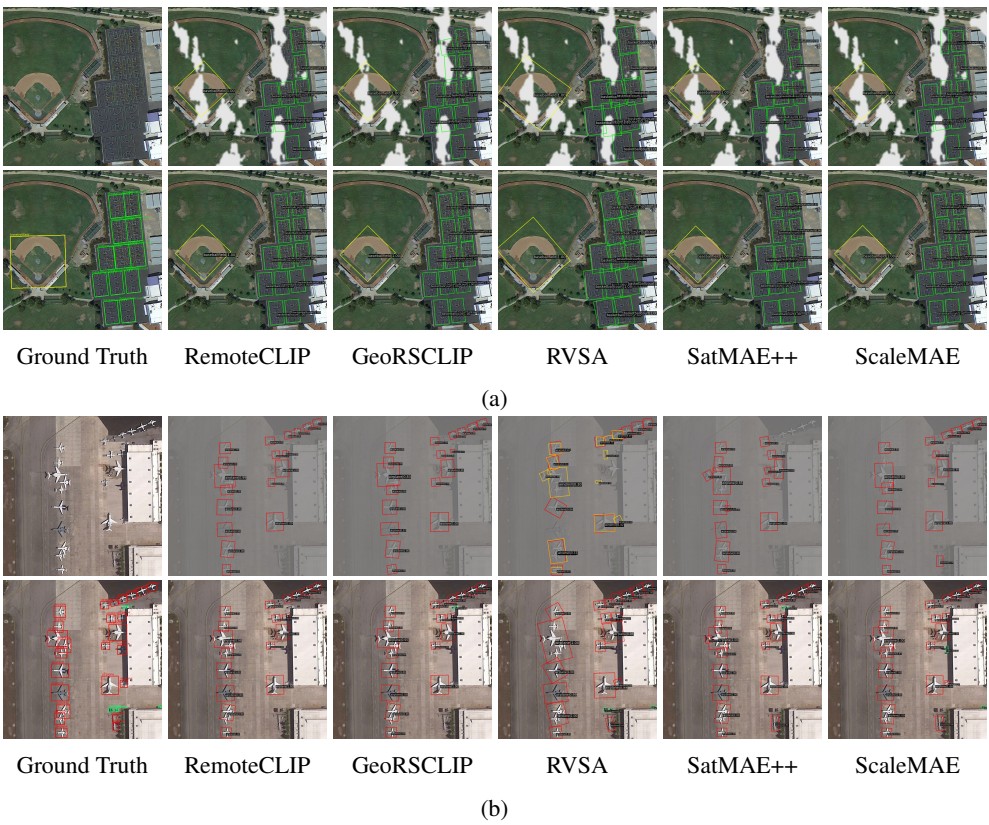

Figure 2: Selected object detection examples under (a) cloud (severity 5) and (b) brightness-and-contrast corruption (severity 5). For each example, the top row shows the corrupted image and the detection results of different models under this corruption, and the bottom row shows the ground truth and detection results on clean images.

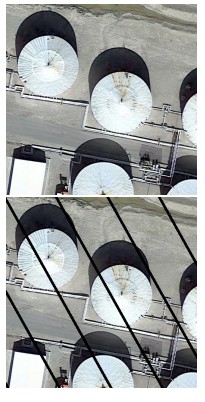

(a)

Top image: 05925_0000.png
Bottom image: **Corruption type:** Data Gaps, **Severity:** 5.

**GT:** The image, sourced from GoogleEarth, shows a facility featuring large storage tanks with distinctive white circular structures against a grey background. There are three visible tanks, arranged from top to bottom within the frame.

**RS-LLaVA:** A factory features numerous storage tanks, with some being white and others having a silver appearance.

**VHM:** The image shows an aerial view of a large oil depot. There are eight large white storage tanks arranged in a square. Each tank has a black ring around the top. There is a road between the tanks and a white building to the left of the tanks.

**Falcon:** the oil tanker is equipped with a large number of oil tanks in the dark. yellow and white neatly placed in the oil storage tank on the ship. three large storage tanks are in a piece of bareland.

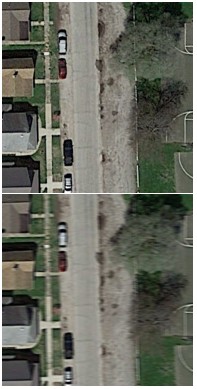

(b)

Top image: 05870_0000.png
Bottom image: **Corruption type:** Motion Blur, **Severity:** 5.

**GT:** Aerial view of a residential area from Google Earth showing a portion of a street with vehicles lined up. The capture shows four parked on the side of the street adjacent to houses. The houses have various roof colors and lawn areas.

**RS-LLaVA:** A basketball court sits on a lawn, surrounded by buildings and trees, with a road nearby.

**VHM:** The image shows a residential area with a tree-lined street. There are houses on both sides of the street. The houses are mostly made of brick and have different colors. The street is made of asphalt and has cars parked on it. There is also a basketball court on the right side of the image.

**Falcon:** a road with two cars and a basketball field is next to a piece of green meadow and several buildings. two cars are running on the wide road and there are some buildings on one side of the road.

Figure 3: Selected image captioning examples. We highlight correct information in green and incorrect information in red.

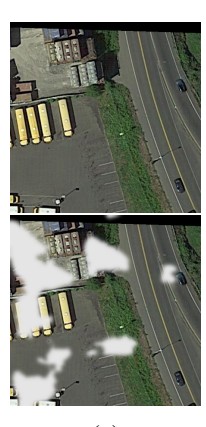

Top image: **P0003_0002.png**
Bottom image: **Corruption type:** Cloud, **Severity:** 5.

*What color are the large vehicles seen in the image?*
**GT**: Yellow, **GeoChat**: Teal, **SkySenseGPT**: teal, **VHM**: yellow.

*How many small vehicles are visible in the image?*
**GT**: 2, **GeoChat**: 11, **SkySenseGPT**: 2, **VHM**: 3.

*Is there a vehicle located at the top-most position in the provided image?*
**GT**: Yes, **GeoChat**: Yes, **SkySenseGPT**: Yes, **VHM**: no.

*What is the orientation of the road in the image?*
**GT**: North-South, **GeoChat**: Right, **SkySenseGPT**: horizontal, **VHM**: north-south.

(a)

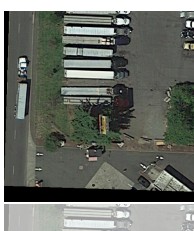

Top image: **P0007_0004.png**
Bottom image: **Corruption type:** Haze, **Severity:** 5.

*What is the predominant color of the large vehicles?*
**GT**: white, **GeoChat**: teal, **SkySenseGPT**: white, **VHM**: white.

*How are the large vehicles predominantly arranged in the parking lot?*
**GT**: in rows, **GeoChat**: Cluster, **SkySenseGPT**: in rows, **VHM**: parked.

*Are there any large vehicles that are positioned away from the main cluster?*
**GT**: Yes, **GeoChat**: Yes, **SkySenseGPT**: Yes, **VHM**: Yes.

*What is the orientation of the large vehicles in the parking lot?*
**GT**: north-south, **GeoChat**: Vertical, **SkySenseGPT**: parallel, **VHM**: parallel.

(b)

Figure 4: Selected VQA examples. Correct answers are shown in green, and incorrect answers are shown in red.

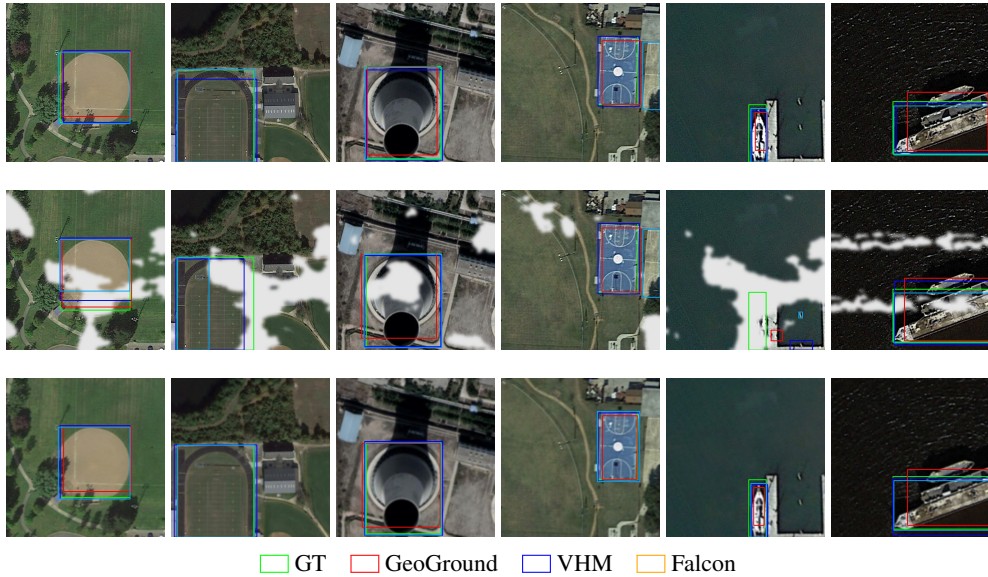

GT  GeoGround  VHM  Falcon

Figure 5: Selected visual grounding examples. The top row presents results on the clean images, while the second and third rows illustrate results under clouds and Gaussian blur corruptions, respectively, both applied with severity level 5. From left to right, the visual grounding questions are: *"The baseball field has a brown infield and is situated in the center of the image surrounded by green grass"*, *"The ground track and field with multiple lanes is located next to a large building with a dark roof"*, *"The chimney seen in the center of the image is cylindrical and extends from the middle to the bottom of the frame"*, *"The basketball court featured in the image is colored in blue with white markings and is located on the right-hand side of the frame"*, *"The small ship is situated vertically near the bottom right corner of the image"*, and *"The ship positioned towards the bottom edge of the image"*.