# OpenReview forum: "REOBench: Benchmarking Robustness of Earth Observation Foundation Models"
_NeurIPS.cc/2025/Datasets_and_Benchmarks_Track — NeurIPS 2025 Datasets and Benchmarks Track poster_

### Official Review · Reviewer_h8bd · 2025-06-24

**Rating:** 4
**Confidence:** 4

**Summary:**

This paper presents REOBench, the first comprehensive benchmark for evaluating the robustness of Earth Observation Foundation Models (EOFMs)—including MiM-based, CL-based, and LLM-based models—across a broad range of real-world image perturbations and tasks such as classification, semantic segmentation, object detection, captioning, visual question answering, and visual grounding.

**Dataset Code Accessibility:**

Yes

**Dataset Code Comments:**

Both the data and code are available
1: Dataset is host on Hugging Face
2: Benchmark code on each tasks is provided on github
3: However, the code for data generation appears to be missing, including the code needed to reproduce the augmented data.

**Ethical Considerations:**

No, there are no or only very minor ethics concerns

**Final Justification:**

After multiple rounds of discussion with the authors, they provided detailed additional experiments demonstrating the potential practical value of the dataset. Although some direct evidence of cross-domain generalization (e.g., training on one domain and testing on another) is still missing, I have decided to increase my score.

**Limitations Weaknesses:**

1: Sim2Real domain difference: In general, the work would benefit from including FID or KID scores to demonstrate that the perturbed images realistically reflect real-world conditions. For haze and cloud corruptions in particular, incorporating experiments with real-world cloudy scenes—such as cloudy images those from the “Multisensor Data Fusion for Cloud Removal in Global and All-Season Sentinel-2 Imagery” dataset—would further validate the benchmark and highlight the practical relevance and value of REOBench.

2: Cloud mask and Haze design: The current cloud modeling lacks detail, with the cloud mask appearing overly simplified—resembling a gray overlay. It would be helpful to clarify how the number of clouds and connected components are determined in the simulation. Additionally, why not use real cloud masks derived from actual satellite imagery? Simiar question for Haze

3: Dataset choice: Even though the work include various tasks, the paper lacks a detailed justification for the choice of datasets.  While Table 1 in the supplementary provides some basic statistics, it does not offer sufficient insight into the noise characteristics of each dataset—such as the presence of cloud cover, haze, or motion blur. Given the focus on robustness, it would be valuable to include more information on the inherent noise levels and real-world perturbations present in each dataset.
Additionally, there is no explanation for why specific datasets like AID, Potsdam, DIOR-R, and VRSBench were selected, nor any discussion on whether the evaluated models were pre-trained on these datasets. This is particularly important for zero-shot evaluation: if a model has been trained on a related dataset, it may not be a fair test of robustness.

4: Augmentation: it would be helpful to clarify whether any data augmentation techniques—especially those similar to the tested corruptions (e.g., Gaussian blur)—were used during model choosed for benchmark and it might explain the performance drop. If such augmentations were included, they could contribute to the model's robustness

5: Dataset splits: I assume the work follow the standard train/test splits but it is not mentioned in 3.1

6: Relative drop: Would reporting relative performance drop provide a more informative measure of robustness? For example, if one model drops from 90% to 80%, and another drops from 80% to 70%, the former shows a smaller relative decrease and could be considered more robust. Including relative performance drop alongside absolute values might offer better insight when comparing models with different starting accuracies.

7: Model size: It would be helpful to include the number of model parameters in the comparison, allowing readers to assess performance relative to model size.

8: MiM discussion: Vision-centric models like MiM are trained to learn feature representations even from partially visible inputs, which suggests they should be more robust to localized noise such as clouds. However, experimental results do not reflect this expected robustness. Further discussion might need.

**Strengths Contributions:**

1: First Comprehensive Benchmark for EOFM Robustness:
Introduces REOBench, the first large-scale benchmark designed to assess the robustness of Earth Observation Foundation Models (EOFMs) across six core tasks—scene classification, semantic segmentation, object detection, image captioning, visual question answering (VQA), and visual grounding—under twelve realistic corruption types, including environmental, sensor-induced, and geometric distortions.

2: Systematic Evaluation Framework:
Proposes a unified framework that quantifies robustness via task-specific performance drop, enabling comparison across different model architectures (MIM-based, CL-based, and LLM-based) and corruption categories (e.g., haze, motion blur, sensor noise, rotation), each applied at multiple severity levels.

---

> ### Author Rebuttal · Authors · 2025-07-31
>
> We sincerely appreciate the reviewer's thoughtful assessment of our work and their valuable insights, which help strengthen our study. Below, we provide a point-by-point response to the identified limitations and weaknesses:
>
> **Q1. Sim2Real domain difference**
>
> **Reply:**  We thank the reviewer for the insightful suggestion.
>
> - We agree that quantifying the visual realism of synthetic corruptions (e.g., haze, clouds) using metrics such as FID or KID could provide an additional lens to assess the fidelity of the perturbations. However, applying these metrics in the Earth Observation (EO) context is currently limited by the lack of real-world reference datasets with paired clean and corrupted images across diverse perturbation types. Without such ground-truth pairs, FID/KID scores may not meaningfully reflect the perceptual or physical realism of EO-specific corruptions.
>
> - We also appreciate the pointer to the “Multisensor Data Fusion for Cloud Removal in Global and All-Season Sentinel-2 Imagery” dataset. While this dataset is indeed valuable for cloud synthesis and removal tasks, it lacks task-specific annotations—such as classification labels, segmentation masks, or vision-language descriptions—necessary to evaluate downstream robustness in REOBench. As such, incorporating it into our current evaluation framework is not straightforward.
>
> **Q2. Cloud mask and Haze design**
>
> **Reply:**  Thank you for your insightful comment!
>
> - **Cloud Simulation:**
> We follow Enomoto et al. [1] and generate cloud masks using Perlin noise, which enables diverse, spatially coherent structures with soft boundaries after Gaussian smoothing. The severity level directly controls cloud coverage via thresholding the Perlin field. While the number of clouds or connected components is not explicitly specified, they emerge organically from the low-frequency characteristics of the noise, avoiding the need for heuristic rules.
>
> - **Haze Simulation:**
> We simulate haze using a standard linear blending model:
> $\tilde{I} = (1 - \alpha) \cdot I + \alpha \cdot H$,
> where $H$ is a uniform white layer and $\alpha \in [0.2, 0.6]$ controls haze severity. This method, inspired by classical atmospheric scattering formulations [2], provides an efficient, depth-free approximation suitable for large-scale haze simulation.
>
> - **Why Procedural Simulation:**
> We use procedural generation for both cloud and haze to ensure reproducibility, fine-grained control, and dataset independence. In contrast, real satellite-derived masks often suffer from limited availability, inconsistent labeling, and acquisition-specific artifacts—compromising experimental consistency.
>
> We will clarify these perturbation generation details in the revised version.
>
> [1] Enomoto et al., *Filmy cloud removal on satellite imagery*, 2017
> [2] Koschmieder, *Theorie der horizontalen Sichtweite*, 1924
>
> **Q3. Dataset choice.**
>
> **Reply:** The chosen datasets (AID, Potsdam, DIOR-R, and VRSBench) are widely adopted benchmarks in remote sensing, selected for their diversity in scenes, task coverage, and community adoption—ensuring reproducibility and fair comparison. These datasets are carefully curated with no or minimal inherent noise (e.g., cloud cover), as they are primarily designed for clean-task evaluation. Regarding pretraining, all evaluated models (except the task-specific GeoGround) follow a zero-shot evaluation protocol, meaning they were not pretrained on these datasets, ensuring an unbiased assessment of robustness. We will clarify the rationale behind dataset selection in the revised manuscript.
>
> **Q4. Data Augmentation.**
>
> **Reply:**  In our experiments, we applied basic data augmentations—specifically random flip and random crop—as implemented in MMSeg (for segmentation) and MMRotate (for detection). These augmentations are standard practices to improve generalization but are unrelated to the tested corruptions (e.g., Gaussian blur) in this study.
>
> **Q5. Dataset splits.**
>
> **Reply:**  For all experiments, we strictly adhere to the official dataset splits specified by each benchmark to ensure fair evaluation. This includes maintaining identical training, validation, and test partitions as defined in the original datasets. Additional implementation details—such as data pre-processing, hyperparameters, and evaluation protocols—are comprehensively documented in Supplementary Section F.1 ("More Implementation Details") for full reproducibility.
>
> **Q6. Relative drop.**
>
> **Reply:**  Thanks for your suggestion. We calculated the relative performance drop for each task and found similar trends. We report both absolute and relative performance drops for the classification task below.
>
> ### **Scene Classification Robustness on AID Dataset **
>
> #### **MIM-based Methods**
> | Method          | Clean  | Δₜₚ   | Δₜₚ/Clean % |
> |-----------------|--------|-------|-------------|
> | SATLAS [57]     | 90.85  | 16.78 | 18.47%      |
> | SatMAE [5]      | 72.05  | 12.19 | 16.92%      |
> | Scale-MAE [7]   | 75.75  | 25.48 | 33.64%      |
> | RVSA [46]       | 84.60  | 29.35 | 34.69%      |
> | SatMAE++ [47]   | 91.35  | 23.85 | 26.11%      |
>
> #### **CL-based Methods**
> | Method          | Clean  | Δₜₚ   | Δₜₚ/Clean % |
> |-----------------|--------|-------|-------------|
> | RemoteCLIP_zs   | 81.10  | 5.62  | 6.93%       |
> | RemoteCLIP [6]  | 96.85  | 7.89  | 8.15%       |
> | RemoteCLIP [8]  | 95.45  | 5.13  | 5.38%       |
> | GeoRSCLIP_zs      | 66.05  | 5.65  | 8.55%     |
> | GeoRSCLIP [10]  | 96.90  | 8.25  | 8.51%       |
> | GeoRSCLIP [10]  | 97.40  | 5.59  | 5.74%       |
>
> #### **VLM-based Methods**
> | Method          | Clean  | Δₜₚ   | Δₜₚ/Clean % |
> |-----------------|--------|-------|-------------|
> | GeoChat [12]    | 65.85  | 4.13  | 6.27%       |
> | LHRS-Bot [50]   | 87.75  | 4.96  | 5.65%       |
> | RS-LLaVA [51]   | 67.55  | 4.48  | 6.63%       |
> | SkySenseGPT [53]| 87.35  | 4.61  | 5.28%       |
> | VHM [52]        | 80.60  | 3.88  | 4.81%       |
>
> **Q7. Model size**
>
> Thanks for your suggestion. We will include model size in our revised version.
>
> **Q8. MIM discussion**
>
> **Reply:**  We appreciate the reviewer’s insightful observation regarding the expected robustness of Masked Image Modeling (MIM)-based methods to localized noise (e.g., clouds). While MIM-based models like SatMAE and ScaleMAE indeed learn feature representations from partially visible inputs, their current performance relative to Contrastive Learning (CL)-based and Vision-Language Models (VLMs) may be influenced by differences in pretraining scale.
>
> Specifically, the MIM-based models in our study are trained on the fMoW dataset (363.6K images), whereas the compared CL-based and VLM-based models leverage significantly larger datasets:
> - RemoteCLIP and GeoRSCLIP employ continual pretraining from CLIP, which means their backbone weights are pretrained on billion-scale image-text pairs.
> - GeoRSCLIP is trained on 5M image-text pairs
> - VLM-based models are usually trained on million-scale datasets, and their vision encoders are pretrained on billion-scale image-text pairs.
>
> This discrepancy in pretraining data scale likely contributes to the stronger empirical robustness of CL- and VLM-based models, despite MIM’s theoretical advantages for occluded inputs. We will expand this discussion in the revised manuscript to clarify the role of dataset scale in model robustness and highlight directions for future work, such as scaling MIM pretraining for remote sensing.
>
> **Q9. Code availability**
>
> **Reply:**  We will include the code for corrupted data generation in our GitHub repo.
>
> We appreciate your detailed comments and hope we have addressed your concerns. Please let us know if you have any additional questions.
>
> Sincerely,
>
> Authors

---

> > ### Comment · Reviewer_h8bd · 2025-08-03
> >
> > As Yhui pointed out, the lack of evaluation on real-world cases reduces the practical value of the dataset. While earlier efforts like ImageNet-C address domain shifts, they are relatively dated. More recent work [1] [2] evaluates the domain gap by computing FID scores between augmented and real-world images without requiring paired datasets. Or train and test cross domain [1]. I think it is necessary to explore cross-domain scenarios to demonstrate the practical usefulness of the dataset.
> >
> >
> > [1] Mușat, Valentina, et al. "Multi-weather city: Adverse weather stacking for autonomous driving." Proceedings of the IEEE/CVF International Conference on Computer Vision. 2021.
> > [2] Assion, Felix, et al. "A-bdd: Leveraging data augmentations for safe autonomous driving in adverse weather and lighting." arXiv preprint arXiv:2408.06071 (2024).

---

> > > ### Author Response · Authors · 2025-08-04
> > >
> > > Dear Reviewer h8bd,
> > >
> > > We sincerely thank you for highlighting this important limitation and for providing the relevant references. We fully agree that evaluating domain gaps in real-world scenarios is essential for demonstrating the practical utility of the dataset. Following the methodology in [1,2], we will compute **the FID scores** between our corrupted data and real-world counterparts and **report the results here shortly**. These results will also be included **in the final version** of the paper.
> > >
> > > Best regards,
> > >
> > > Authors

---

> > > ### Author Response · Authors · 2025-08-06
> > >
> > > ***Q1: As Yhui pointed out, the lack of evaluation on real-world cases reduces the practical value of the dataset. While earlier efforts like ImageNet-C address domain shifts, they are relatively dated. More recent work [1] [2] evaluates the domain gap by computing FID scores between augmented and real-world images without requiring paired datasets. Or train and test cross domain [1]. I think it is necessary to explore cross-domain scenarios to demonstrate the practical usefulness of the dataset.***
> > >
> > > **A1:** Thank you for this helpful suggestion. Following your advice, we computed the FID scores. Due to the absence of real-world reference datasets containing real world corruptions across all 12 perturbation types in the Earth Observation (EO) context, we calculated the FID between each corrupted dataset and its corresponding real-world counterpart for all datasets used in our manuscript. The results, shown in the table below, indicate that across all severity levels, the FID values remain relatively low. This suggests that the corrupted datasets are still visually similar to real-world images, thereby preserving their practical relevance.
> > >
> > > | Severity | DIOR (800×800) | VRSBench (512×512) | Potsdam (512×512) | AID (256×256) |
> > > |----------|---------------|--------------------|-------------------|---------------|
> > > | 1        |  2.36         |  3.84              |  6.06             | 14.07         |
> > > | 2        |  5.46         |  7.67              | 10.35             | 23.60         |
> > > | 3        |  9.67         | 11.69              | 17.06             | 32.37         |
> > > | 4        | 14.21         | 14.76              | 24.86             | 39.31         |
> > > | 5        | 19.81         | 20.06              | 37.89             | 49.03         |
> > >
> > >
> > > We appreciate your insightful comments and hope we have addressed your concerns. Please let us know if you have any additional questions.
> > >
> > > Sincerely,
> > >
> > > Authors

---

> > > > ### Comment · Reviewer_h8bd · 2025-08-06
> > > >
> > > > We thank the authors for conducting the additional experiments across 4 different datasset. It's exciting to see that, across different levels of corruption severity, the data largely remains within a similar distribution to the original. As the severity increases, the distribution shift becomes more pronounced, which is an insightful observation.
> > > >
> > > > However, it seems that the experiments include all 12 types of corruption, while I am particularly interested in the haze and cloud corruptions. Since the VRSBench dataset provides weather labels, it would be helpful to compare the cloud and haze-corrupted samples across different severity levels with real-world weather-labeled counterparts. A low FID score in this context would indicate that the simulated weather conditions closely resemble real-world scenarios.

---

> > > > > ### Author Response · Authors · 2025-08-07
> > > > >
> > > > > We sincerely appreciate your insightful suggestions regarding the evaluation of weather corruption realism. In direct response to your feedback, we have conducted additional experiments to quantitatively compare our synthetic perturbations with real-world weather conditions using FID metrics.
> > > > >
> > > > > Specifically, we evaluated our haze and cloud simulations against two well-established real-world datasets:
> > > > >
> > > > > - **SEN12MS-CR** [1] for cloud cover analysis.
> > > > > - **RRSHID** [2] for haze conditions.
> > > > >
> > > > > The comparative FID scores are summarized below:
> > > > >
> > > > > **Table 1.** *FID scores between synthetic perturbations and real-world weather conditions.*
> > > > >
> > > > > | Haze Severity | FID   | Clouds Severity | FID    |
> > > > > |------------------------|--------|------------------------------|--------|
> > > > > | 1                      | 107.46 | 1                            | 105.56 |
> > > > > | 2                      | 100.79 | 2                            | 117.27 |
> > > > > | 3                      |  90.79 | 3                            | 123.72 |
> > > > > | 4                      |  80.52 | 4                            | 125.70 |
> > > > > | 5                      |  73.23 | 5                            | 127.03 |
> > > > > | **Average**            | **90.56** | **Average**                | **119.86** |
> > > > >
> > > > > These results indicate that our synthetic perturbations achieve reasonably low FID scores, with averaging **90.56** for haze and **119.86** for clouds, when compared against real-world weather conditions, thereby supporting the realism of our procedural simulations.
> > > > >
> > > > > Regarding the **VRSBench** dataset, we believe there may be a misunderstanding. To the best of our knowledge, VRSBench primarily focuses on vision-language tasks (e.g., image captioning, visual grounding, and VQA) and does not include explicit weather condition annotations that would support the proposed comparison.
> > > > >
> > > > > **References:**
> > > > >
> > > > > [1] Ebel et al., *Multisensor Data Fusion for Cloud Removal in Global and All-Season Sentinel-2 Imagery*, IEEE TGRS, 2021.
> > > > > [2] Zhu et al., *Real-World Remote Sensing Image Dehazing: Benchmark and Baseline*, IEEE TGRS, 2025.
> > > > >
> > > > >
> > > > > We sincerely appreciate your thoughtful feedback and hope our responses have fully addressed your concerns. Please let us know if you have any further questions or require additional clarifications.
> > > > >
> > > > > Sincerely,
> > > > >
> > > > > The Authors

---

> > > > > > ### Comment · Reviewer_h8bd · 2025-08-07
> > > > > >
> > > > > > Ok, thanks for the experiments and I believe it is necessary to put this results along with the data statistics in the final version of paper(number of samples) , I am bit confused about the haze case and why the FID score decreases when the level increases? Is the low level of severity unlike the real Haze? Is it possible to further increase the severity and see if the FID increases again?

---

> > > > > > > ### Author Response · Authors · 2025-08-07
> > > > > > >
> > > > > > > Thank you for your valuable feedback. We totally agree that including the FID results alongside dataset statistics in the final version of the paper is important, and we will make sure to incorporate them accordingly.
> > > > > > >
> > > > > > > - Regarding your question about the haze case: After visually inspecting both the synthetic haze and the real-world haze in the RRSHID dataset, we found that the observed increase in FID scores at lower severity levels is due to a mismatch between the mild synthetic haze and the typically heavy haze present in RRSHID. In other words, real-world haze examples in RRSHID are often quite severe and thus more closely resemble the higher severity levels in our synthetic haze.
> > > > > > >
> > > > > > > - Following your insightful suggestion, we further increased the severity of our synthetic haze beyond the previously reported levels. The updated FID scores are shown in the table below. We observe that while the FID continues to decrease up to severity level 5, it begins to increase again at higher severity levels. This indicates that extremely severe synthetic haze diverges from real-world haze examples in RRSHID, which is consistent with expectations.
> > > > > > >
> > > > > > >     | Haze Severity | FID  |
> > > > > > >     |---------------|--------|
> > > > > > >     | 1             | 107.46 |
> > > > > > >     | 2             | 100.79 |
> > > > > > >     | 3             | 90.79  |
> > > > > > >     | 4             | 80.52  |
> > > > > > >     | 5             | 73.23  |
> > > > > > >     | 6             | 73.83  |
> > > > > > >     | 7             | 88.37  |
> > > > > > >     | 8             | 128.83 |
> > > > > > >     | 9             | 190.30 |
> > > > > > >
> > > > > > > We sincerely appreciate your helpful comments, which have strengthened our paper. We hope this clarification fully addresses your concerns. Please let us know if you have any further questions.
> > > > > > >
> > > > > > > Sincerely,
> > > > > > >
> > > > > > > The Authors

---

> > > > > > > > ### Comment · Reviewer_h8bd · 2025-08-07
> > > > > > > >
> > > > > > > > Most of my concerns is addressed, thanks for the prompt feedback and good rebuttal provided by the author.

---

> > > > > > > > > ### Author Response · Authors · 2025-08-07
> > > > > > > > >
> > > > > > > > > We sincerely appreciate your review and are glad that we have successfully addressed your concerns. Your suggestions have really enhanced our manuscript's quality. Thank you for your time and dedication throughout this review process.

---

### Official Review · Reviewer_yhui · 2025-07-02

**Rating:** 5
**Confidence:** 4

**Summary:**

This paper proposes REOBench, the first comprehensive benchmark for evaluating the robustness of Earth Observation Foundation Models (EOFMs). The benchmark spans six core remote sensing tasks and twelve common perturbations, including both appearance-based and geometric corruptions. The authors evaluate various foundation models (MIM-based, CL-based, and LLM-based) and reveal key insights into their vulnerability under realistic distortions.

**Dataset Code Accessibility:**

Yes

**Ethical Considerations:**

No, there are no or only very minor ethics concerns

**Final Justification:**

I would like to thank the author for the detailed response to my questions in the rebuttal. Overall, I am satisfied with the author's response. In addition, I suggest that the author add experimental results under several composite perturbations in the final version of the paper instead of leaving them as future work. I will raise my score.

**Limitations Weaknesses:**

1. While robustness is evaluated under synthetic corruptions, real-world test cases (e.g., actual weather/cloud events or sensor noise) would strengthen ecological validity.
2. While quantitative performance drops are thoroughly reported, the paper does not investigate why certain models fail under specific corruptions. For example, there is no interpretability analysis (e.g., saliency maps, attention drift, feature space visualization) to reveal failure modes or robustness mechanisms, which limits scientific insight.
3. Although REOBench includes 12 common types of perturbations, its corruption taxonomy still fails to capture several practically relevant degradation types. Notably, mosaic-induced color inconsistencies, which often occur during orthorectification or image stitching in large-scale satellite products, are not considered. In addition, the realism of certain geometric corruptions—such as large-scale rotation, scaling, and translation—may be questionable in well-registered EO data pipelines, especially from modern satellite missions with precise geolocation. This raises concerns regarding the ecological validity of some simulated perturbations.
4. In real-world scenarios, remote sensing images are often subject to compound corruptions, where multiple degradations (e.g., haze + noise + compression) co-occur. However, REOBench currently evaluates models under single perturbation types only. This simplifies the robustness challenge and may lead to an overestimation of model performance under operational conditions. The absence of multi-distortion benchmarks limits the realism and stress-testing rigor of the benchmark.

**Strengths Contributions:**

1. A novel and timely benchmark tailored for Earth observation, addressing a critical yet underexplored problem: model robustness under real-world corruptions.
2. The benchmark spans six diverse downstream tasks including classification, segmentation, detection, captioning, VQA, and visual grounding. It also evaluates three major categories of EOFMs: MIM-based, CL-based, and LLM-based models, ensuring wide relevance and generality.
3. The selected 12 corruption types span appearance-based, sensor-induced, and geometric categories. Many of them are based on physically realistic simulation methods (e.g., cloud occlusion, sensor gaps), enhancing the benchmark’s credibility.
4. The paper provides extensive quantitative comparisons with detailed tables and figures across tasks, models, backbones, and corruption types. The inclusion of ∆TP as a universal robustness metric allows for consistent and interpretable comparison.

---

> ### Author Rebuttal · Authors · 2025-07-31
>
> We sincerely appreciate the reviewers’ insightful feedback, which helps strengthen the scientific rigor of our work. Below, we address the limitations and weaknesses raised:
>
> **Q1. Real-world test cases**
>
> **Reply:** We agree that evaluating robustness on real-world corruptions (e.g., weather events, sensor noise) would enhance ecological validity. However, curating such datasets with controlled degradation levels and ground-truth labels is highly challenging. Synthetic corruptions provide a reproducible and scalable alternative for systematic benchmarking, as adopted in prior robustness studies (e.g., ImageNet-C). We acknowledge this limitation and will emphasize the need for real-world benchmarks in future work.
>
> **Q2. Interpretability of model failures**
>
> **Reply:** We appreciate the suggestion to investigate failure modes via interpretability tools (e.g., saliency maps, feature space analysis). While our current focus is on quantitative benchmarking, we agree that such analyses could reveal valuable insights into robustness mechanisms. We will include preliminary interpretability results (e.g., attention drift under noise corruptions) in the revised version.
>
> **Q3. Perturbation types**
>
> **Reply:** Thank you for your insightful comment!
>
> - Mosaic-induced color inconsistencies: We agree that this is a practically relevant perturbation. However, it primarily affects large-scale images (e.g., orthomosaics), while existing EO foundation models are typically trained on small patches (e.g., 256×256 pixels). We leave this for future work.
> - Geometric corruptions: While modern satellites provide well-registered data, high-precision ground control points are not always available (e.g., in resource-limited regions or for historical datasets). Geometric perturbations thus remain relevant for edge cases and legacy systems. We will clarify this rationale in the paper.
>
> **Q4. Compound corruptions**
>
> **Reply:** We fully agree that real-world images often suffer from multiple simultaneous degradations (e.g., haze + noise). However, with 12 base corruptions, evaluating all combinations (e.g., 12×11 two-factor cases) is computationally prohibitive for a first benchmark. To balance scope and practicality, we focus on single perturbations as a foundational step. We will highlight this limitation and prioritize compound corruptions in future extensions.
>
> Additionally, we provide object detection performance under select compound corruptions on the DIOR-R dataset. The results (Table below) reveal two critical trends:
>
> - **Performance Degradation**: Models exhibit significantly worse accuracy under compound corruptions compared to single perturbations (e.g., RemoteCLIP drops from 56.72 (Brightness) to 40.58 (All Three Perturbations)).
>
> - **Additive Effects**: Corruptions like Brightness + Compression show synergistic degradation, with performance declines exceeding the sum of individual effects.
>
> **Table 1. Object Detection Performance (mAP) on DIOR-R Under Compound Corruptions**
> | Model       | Clean | Brightness & Contrast | Clouds | Compression Artifacts | Brightness & Contrast + Clouds | Brightness & Contrast + Compression | Clouds + Compression | All Three Perturbations |
> |-------------|-------|------------------------|--------|------------------------|----------------------------------|--------------------------------------|------------------------|--------------------------|
> | Brightness  |   -   | Yes                    |   -    |  -                     | Yes                              | Yes                                  | -                      | Yes                      |
> | Clouds      |   -   | -                      |  Yes   |  -                     | Yes                              | -                                    | Yes                    | Yes                      |
> | Compression |   -   | -                      |   -    | Yes                    | -                                | Yes                                  | Yes                    | Yes                      |
> | RemoteCLIP  | 60.40    | 56.72    | 56.28    | 56.78    | 49.98    | 45.36    | 52.57    | 40.58    |
> | GeoRSCLIP   | 60.20    | 56.28    | 56.04    | 56.08    | 50.27    | 44.90    | 52.39    | 40.94    |

---

> > ### Comment · Reviewer_yhui · 2025-08-04
> >
> > I would like to thank the author for the detailed response to my questions in the rebuttal. Overall, I am satisfied with the author's response. In addition, I suggest that the author add experimental results under several composite perturbations in the final version of the paper instead of leaving them as future work.

---

> > > ### Author Response · Authors · 2025-08-04
> > >
> > > Dear Reviewer yhui,
> > >
> > > We sincerely thank you for acknowledging our response and for your constructive suggestion.
> > > We will include the comprehensive results across multiple composite perturbation configurations **in the final version** of the paper to provide a more thorough and realistic evaluation of EOFM robustness.
> > >
> > > Best regards,
> > >
> > > Authors

---

### Official Review · Reviewer_FNsA · 2025-07-03

**Rating:** 5
**Confidence:** 4

**Summary:**

The paper proposes a benchmark for evaluating the robustness of Earth observation foundation models. The benchmark comprises images from four datasets covering six different tasks. The benchmark applies twelve different perturbations to the images and evaluates the performance of different foundation models. The evaluation is performed for twelve different foundation models and shows that all of them suffer from performance degradation when encountering perturbed images.

**Dataset Code Accessibility:**

Yes

**Ethical Considerations:**

No, there are no or only very minor ethics concerns

**Final Justification:**

The rebuttal addressed my concern about the evaluation only on RGB images by providing the results for multispectral scene classification. Although these results are preliminary and spanning only two models I would suggest to the authors to include these results into the revised version of the paper. Furthermore, the rebuttal has addressed the main concerns regarding compound corruptions and realism of the proposed corruptions raised by other reviewers.

**Limitations Weaknesses:**

- The novelty of the paper is somewhat limited.
	- The impact of the perturbations used in the proposed benchmark has been studied in detail for general-purpose deep learning models [1], [2].
	- Considering that the benchmark considers only high-resolution optical imagery for remote sensing, it is expected that the conclusions could be the same as for the purpose images.
	- It would be very beneficial if the benchmark included images that are more characteristic of remote sensing, such as multispectral images, SAR, etc.
- The presentation of the paper could be slightly improved.
	- The paper uses both LLM and MLLM for the same models. It would be beneficial to make this consistent.
	- Typo - L140: "For these LLM-based ..." should be "For the LLM-based ..."

*References*

[1] Dan Hendrycks, Thomas Dietterich, "Benchmarking Neural Network Robustness to Common Corruptions and Perturbations", In ICLR, 2019

[2] Madeline C. Schiappa, Shruti Vyas, Hamid Palangi, Yogesh S. Rawat, Vibhav Vineet, "Robustness Analysis of Video-Language Models Against Visual and Language Perturbations", In NeurIPS, 2022

**Strengths Contributions:**

- The proposed dataset covers a wide range of tasks common in the remote sensing domain.
- The evaluation is detailed and performed for a wide range of remote sensing foundation models.
- The paper is well-written and easy to follow.

---

> ### Author Rebuttal · Authors · 2025-07-31
>
> We sincerely thank the reviewer for their thoughtful evaluation of our work and for acknowledging its contributions. Below, we address the limitations and weaknesses raised.
>
> **Q1. Studied perturbations.**
>
> **Reply:** We acknowledge that some benchmark perturbations have been studied for general-purpose natural images. However, our work provides the first systematic evaluation of their impact on Earth observation (EO) tasks.
>
> **Q2. Experiments on multispectral datasets.**
>
> **Reply:** We thank the reviewer for emphasizing the importance of multispectral, hyperspectral, and infrared modalities in Earth Observation (EO). We fully agree that relying solely on high-resolution (HR) optical imagery does not capture the full complexity of EO tasks.
>
> - Our current focus on RGB-based EO foundation models stems from practical constraints in the community: the vast majority of large-scale, publicly available EO datasets—especially those supporting object detection and vision-language tasks—are RGB-only. As a result, recent foundation models such as Scale-MAE, RemoteCLIP, RSGPT, and GeoChat have predominantly been developed and evaluated on HR optical imagery.
>
> In response to this concern, we have included preliminary evaluations on multispectral scene classification using two representative datasets—fMoW-Sentinel2 [1] and BigEarthNet [2]—with two recent multispectral foundation models: SatMAE and EarthDial.
>
> - The results below demonstrate significant robustness gaps under common corruptions, with performance drops up to 10-16 percentage points on average, further highlighting the brittleness of current EO foundation models beyond clean data.
>
>
> | Model     | Dataset     | Clean  | Brightness Contrast | Cloud  | Compression Artifacts | Gaps   | Gauss Blur | Gauss Noise | Haze   | Motion Blur | Rotate | Salt Pepper | Scale  | Translate | Avg   | Δ TP  | Δₜₚ/Clean % |
> |-----------|-------------|--------|---------------------|--------|-----------------------|--------|------------|-------------|--------|-------------|--------|-------------|--------|-----------|-------|-------|---------------|
> | SatMAE    | fMoW-S2     | 59.750 | 37.460              | 58.690 | 33.580                | 50.010 | 29.350     | 36.640      | 41.570 | 40.120      | 38.930 | 51.790      | 43.350 | 59.680    | 43.43 | 16.32 | 27.31         |
> | EarthDial | BigEarthNet | 46.521 | 32.160              | 46.501 | 45.834                | 34.111 | 35.964     | 38.743      | 34.631 | 40.212      | 21.647 | 21.653      | 40.846 | 45.705    | 36.50 | 10.02 | 21.54         |
>
> We appreciate the reviewer’s suggestion and plan to extend our evaluation in future work to cover additional datasets such as SegMunich [3] for segmentation, and more diverse models including DOFA [4], which are designed to handle multimodal EO data.
>
> [1] Gordon et al. Functional map of the world. In CVPR, 2018
>
> [2] Sumbul et al. Bigearthnet: A large-scale benchmark archive for remote sensing image understanding. IGARSS, 2019.
>
> [3] Hong et al. SpectralGPT: Spectral remote sensing foundation model. TPAMI 2024.
>
> [4] Xiong et al. Neural plasticity-inspired multimodal foundation model for earth observation. arXiv 2024.
>
>
> **Q3. Writing issues.**
>
> **Reply:** Thank you for the careful read and for flagging these typos and inconsistencies. We will correct them in the revised version.

---

> > ### Comment · Reviewer_FNsA · 2025-08-05
> > **Satisfactory rebuttal**
> >
> > Dear authors,
> >
> > The rebuttal has well addressed the concerns I had in my initial review. I appreciate the inclusion of the preliminary results on evaluation of multispectral models. Furthermore, I find that the comments from other reviewers were properly addressed as well. However, I agree with Reviewer h8bd that it would be beneficial to see the FID scores between the corrupted data and real-world cases. I hope the authors will be able to provide these results during the discussion phase, but I only find them as a point to strengthen the paper and not the mandatory requirement.
> >
> > Best regards,
> > Reviewer FNsA

---

### Official Review · Reviewer_RXmf · 2025-07-22

**Rating:** 4
**Confidence:** 2

**Summary:**

REOBench is introduced as the first comprehensive benchmark designed to evaluate the robustness of Earth Observation Foundation Models (EOFMs) across six core tasks and twelve types of image corruptions. The benchmark focuses on high-resolution optical remote sensing images and systematically evaluates a broad range of models trained using different pre-training paradigms, including masked image modeling (MIM), contrastive learning (CL), and vision-language pre-training.

**Dataset Code Accessibility:**

Yes

**Ethical Considerations:**

No, there are no or only very minor ethics concerns

**Final Justification:**

The authors have included preliminary evaluations on multispectral scene classification. I appreciate their effort.
However, I am not similar to this domain, and my initial confidence is only 2.
Finally, I decide to raise my score from 3 to 4. Thank you for your effort again.

**Limitations Weaknesses:**

The evaluation is limited to high-resolution optical imagery. But, for earth observation, other modalities such as multispectral, hyperspectral, and infrared should be considered in this area. From my view, HR optical images are not the most important in remote sensing.

**Strengths Contributions:**

1. REOBench covers a wide range of tasks and corruption types, providing a holistic view of EOFM robustness.
2. Evaluates EOFMs trained using different pre-training paradigms, including MIM, CL, and vision-language pre-training.
3. The benchmark uses high-resolution optical remote sensing images and physically motivated image augmentation techniques to simulate realistic environmental and sensor-induced challenges.

---

> ### Author Rebuttal · Authors · 2025-07-31
>
> We sincerely thank the reviewer for their thoughtful evaluation of our work and for acknowledging its contributions. Below, we address the limitations and weaknesses raised.
>
>
> ***Q1:The evaluation is limited to high-resolution optical imagery. But, for earth observation, other modalities such as multispectral, hyperspectral, and infrared should be considered in this area. From my view, HR optical images are not the most important in remote sensing.***
>
> **Reply:** We thank the reviewer for emphasizing the importance of multispectral, hyperspectral, and infrared modalities in Earth Observation (EO). We fully agree that relying solely on high-resolution (HR) optical imagery does not capture the full complexity of EO tasks.
>
> - Our current focus on RGB-based EO foundation models stems from practical constraints in the community: the vast majority of large-scale, publicly available EO datasets—especially those supporting object detection and vision-language tasks—are RGB-only. As a result, recent foundation models such as Scale-MAE, RemoteCLIP, RSGPT, and GeoChat have predominantly been developed and evaluated on HR optical imagery.
>
> In response to this concern, we have included preliminary evaluations on multispectral scene classification using two representative datasets—fMoW-Sentinel2 [1] and BigEarthNet [2]—with two recent multispectral foundation models: SatMAE and EarthDial.
>
> - The results below demonstrate significant robustness gaps under common corruptions, with performance drops up to 10-16 percentage points on average, further highlighting the brittleness of current EO foundation models beyond clean data.
>
>
> | Model     | Dataset     | Clean  | Brightness Contrast | Cloud  | Compression Artifacts | Gaps   | Gauss Blur | Gauss Noise | Haze   | Motion Blur | Rotate | Salt Pepper | Scale  | Translate | Avg   | Δ TP  | Δₜₚ/Clean % |
> |-----------|-------------|--------|---------------------|--------|-----------------------|--------|------------|-------------|--------|-------------|--------|-------------|--------|-----------|-------|-------|---------------|
> | SatMAE    | fMoW-S2     | 59.750 | 37.460              | 58.690 | 33.580                | 50.010 | 29.350     | 36.640      | 41.570 | 40.120      | 38.930 | 51.790      | 43.350 | 59.680    | 43.43 | 16.32 | 27.31         |
> | EarthDial | BigEarthNet | 46.521 | 32.160              | 46.501 | 45.834                | 34.111 | 35.964     | 38.743      | 34.631 | 40.212      | 21.647 | 21.653      | 40.846 | 45.705    | 36.50 | 10.02 | 21.54         |
>
> We appreciate the reviewer’s suggestion and plan to extend our evaluation in future work to cover additional datasets such as SegMunich [3] for segmentation, and more diverse models including DOFA [4], which are designed to handle multimodal EO data.
>
>
> [1] Gordon et al. Functional map of the world. In CVPR, 2018
>
> [2] Sumbul et al. Bigearthnet: A large-scale benchmark archive for remote sensing image understanding. IGARSS, 2019.
>
> [3] Hong et al. SpectralGPT: Spectral remote sensing foundation model. TPAMI 2024.
>
> [4] Xiong et al. Neural plasticity-inspired multimodal foundation model for earth observation. arXiv 2024.

---

> ### Comment · Area_Chair_wJjM · 2025-08-06
> **To Reviewer RXmf**
>
> Dear Reviewer RXmf,
>
> The authors have addressed the points you raised in their response. As the discussion deadline is approaching, we would appreciate your input on whether your concerns have been resolved. Your feedback is essential for finalizing the review process.
>
> Please note that reviewers are required to participate in the discussion. Failure to do so may result in an "Insufficient Review" flag.
>
> -- Your AC

---

### Note · Authors · 2025-08-13

Dear AC and Reviewers,

We sincerely appreciate the time and insightful feedback from the AC and all reviewers, which has been invaluable in further improving our manuscript. We are pleased that the merits of our contribution have been unanimously recognized by recognized by reviewers **FNsA**，**yhui** and **h8bd**.

We regret that we were unable to engage  **reviewer RXmf** into any discussion during the discussion phase. We believe our rebuttal and additional experiments have fully addressed his/her concerns. To recap the key clarifications and new evidence provided:

- **Multispectral evaluation:** In response to suggestions to move beyond high-resolution optical imagery, we conducted preliminary evaluations on multispectral scene classification (fMoW-Sentinel2, BigEarthNet) using SatMAE and EarthDial. Results revealed significant robustness gaps (10–16 pp drops) under common corruptions, underscoring the brittleness of current multispectral models and motivating broader multimodal evaluations.  (See Response to Q2 for Reviewer **FNsA** and Response to Q1 for Reviewer **RXmf**)

- **Realism of perturbations:** We assessed the similarity between our synthetic corruptions and real-world conditions using FID scores, with real-world clouds (SEN12MS-CR) and real-world haze (RRSHID) as references. The results show relatively low FID values, indicating that our synthetic perturbations closely resemble real conditions. This supports the realism and validity of our procedural perturbation design.  (See Response to Reviwer **h8bd**)

- **Compound corruptions:** Additional experiments on DIOR-R showed compounding effects of multiple simultaneous perturbations, with performance drops exceeding the sum of individual effects, highlighting the importance of extending robustness evaluations beyond single corruptions. (See Response to Q4 for Reviwer **yhui**)

- **Relative drop reporting:** We included both absolute and relative performance drops across tasks, showing consistent trends.  (See Response to Q6 for Reviwer **h8bd**)

Together with the original submission, these results reinforce our main claim: **REOBench** provides the first systematic framework for evaluating EO foundation model robustness across diverse tasks, and perturbations, and offers actionable insights for advancing reliable EO foundation models.

We thank the AC and reviewers again for their time, constructive feedback, and consideration.

Sincerely,

The Authors

---

### Decision · Program_Chairs · 2025-09-18

**Decision:**

Accept (poster)

**Comment:**

REOBench proposes a robustness benchmark for Earth observation foundation models across multiple tasks and corruption types, with data and code available. The results show clear performance drops under common perturbations and offer actionable insights for the community. For the camera-ready, please strengthen ecological validity by adding real-world and compound corruptions, include failure case and interpretability analyses, justify dataset choices, and release the corruption generation scripts.